# Biophysics of object segmentation in a collision-detecting neuron

Richard Burkett Dewell[1], Fabrizio Gabbiani[1,2]*

[1]Department of Neuroscience, Baylor College of Medicine, Houston, United States; [2]Electrical and Computer Engineering, Rice University, Houston, United States

**Abstract** Collision avoidance is critical for survival, including in humans, and many species possess visual neurons exquisitely sensitive to objects approaching on a collision course. Here, we demonstrate that a collision-detecting neuron can detect the spatial coherence of a simulated impending object, thereby carrying out a computation akin to object segmentation critical for proper escape behavior. At the cellular level, object segmentation relies on a precise selection of the spatiotemporal pattern of synaptic inputs by dendritic membrane potential-activated channels. One channel type linked to dendritic computations in many neural systems, the hyperpolarization-activated cation channel, HCN, plays a central role in this computation. Pharmacological block of HCN channels abolishes the neuron's spatial selectivity and impairs the generation of visually guided escape behaviors, making it directly relevant to survival. Additionally, our results suggest that the interaction of HCN and inactivating $K^+$ channels within active dendrites produces neuronal and behavioral object specificity by discriminating between complex spatiotemporal synaptic activation patterns.

DOI: https://doi.org/10.7554/eLife.34238.001

## Introduction

Neurons within the brain receive information about the outside world through a continuous ever-changing stream of synaptic inputs. These inputs can arrive thousands of times a second spread out across tens or even hundreds of thousands of different synaptic locations. Ultimately, the primary task of a neuron is to filter out the irrelevant elements of this dynamic stream and extract from the noisy cascade features meaningful for the animal. While the importance of timing of synaptic inputs is well known, the role of the spatial pattern of dendritic inputs has received less attention. In fact, it is still an unsettled question whether neurons extract information embedded within the broader spatial patterns of ongoing synaptic inputs (*Grienberger et al., 2015*).

In support of this hypothesis, recent investigations demonstrate spatial patterning of excitatory and inhibitory synaptic inputs (*Wilms and Häusser, 2015*; *Bloss et al., 2016*; *Gökçe et al., 2016*; *Bloss et al., 2018*) and dendritic processes capable of discriminating between different such patterns (*Smith et al., 2013*; *Weber et al., 2016*; *Wilson et al., 2016*). For instance, local synaptic clustering produces supralinear summation which enhances the selectivity of visual neurons (*Smith et al., 2013*; *Wilson et al., 2016*). Studies of clustering have focused on fast positive feedback, such as the dendritic spikes and NMDA receptors that amplify local patterns of synaptic inputs, thereby conferring directional selectivity to some retinal ganglion cells (*Sivyer and Williams, 2013*; *Poleg-Polsky and Diamond, 2016*). Recent results also illustrate the functional role of fine scale synaptic patterning in many neurons (*Druckmann et al., 2014*; *Kleindienst et al., 2011*; *Petreanu et al., 2009*; *Takahashi et al., 2012*), but whether neurons also discriminate between broad spatiotemporal patterns embedded across thousands of synaptic inputs remains largely unknown. As many neuron types receive an ongoing stream of tens of thousands of inputs spread across a dendritic arbor, the ability to discriminate between such synaptic patterns would markedly

**\*For correspondence:**
gabbiani@bcm.edu

**Competing interests:** The authors declare that no competing interests exist.

**eLife digest** Whether you are a flying insect or a driver on a freeway, your survival will depend on avoiding collisions. Many species have nerve cells or neurons within their visual system that respond to objects headed towards them on a collision course. In locusts, for example, a neuron called the lobula giant movement detector (LGMD) triggers an escape response upon detecting an impending collision. But how does it do this?

The answer may also help us understand how neurons process other complex inputs. This is because like most neurons, the LGMD contains thousands of branches called dendrites. The job of the dendrites is to receive input from other neurons and collect that input for processing. In the LGMD's case, each piece of input reveals what is happening at a single point in the locust's field of vision. The cell combines all the inputs and uses the end result to decide whether to trigger an escape response.

An object on a collision course will generate a specific sequence of images in the locust's eye. These images will activate LGMD dendrites in a specific pattern. To test whether LGMD neurons use this pattern to detect approaching objects, Dewell and Gabbiani showed locusts two sets of movies. One set featured an object looming towards the insect on a collision course. But in the other set, the same movies had been scrambled. These movies thus activated LGMD dendrites in a different pattern than the movies showing looming objects. Both the LGMD neurons, and the locusts themselves, responded more to the non-scrambled movies. This suggests that they do use the pattern of activity in dendrites to detect impending collisions.

Blocking two types of ion channels in the membrane of the dendrites prevented the neurons from distinguishing between scrambled and non-scrambled movies. Both of these ion channels are also present in the dendrites in our own brains. This suggests that many neurons can detect the spatial pattern in which their dendrites become active. By revealing how neurons process complex visual inputs, the results of Dewell and Gabbiani could help improve algorithms for man-made collision avoidance systems. These could be used in self-driving cars, or in technology to help visually impaired people navigate independently.

DOI: https://doi.org/10.7554/eLife.34238.002

increase their computational power. Additionally, a neuron's computational task likely determines which aspects of the spatiotemporal pattern of synaptic activities are most relevant and constrains the nonlinear dynamics of the membrane potential in its dendrites (*Spruston, 2008*; *Ujfalussy et al., 2015*). To address these issues, we focus on large-scale processing of synaptic inputs and the dendritic computations required for visual object segmentation in the context of collision avoidance behaviors.

The spatiotemporal sequence of synaptic inputs relevant to collision avoidance is determined by the statistics of the approaching object. Objects approaching on a collision course or their simulation on a screen, called looming stimuli, produce a characteristic visual stimulus on the observer's retina, expanding coherently in all directions with increasing angular velocity. Discriminating this retinal pattern from that of optic flow or from that of an object approaching on a miss trajectory requires integrating information across many points in time and space. Among neurons capable of such discrimination (*Sun and Frost, 1998*; *Nakagawa and Hongjian, 2010*; *Liu et al., 2011*; *de Vries and Clandinin, 2012*; *Dunn et al., 2016*; *Klapoetke et al., 2017*), the lobula giant movement detector (LGMD, *Figure 1A*) has been extensively studied: it is an identified neuron of the grasshopper optic lobe located three synapses away from photoreceptors (*O'Shea and Williams, 1974*). The LGMD responds maximally to looming stimuli (*Schlotterer, 1977*; *Rind and Simmons, 1992*; *Hatsopoulos et al., 1995*) with a characteristic firing rate profile (*Hatsopoulos et al., 1995*; *Gabbiani et al., 1999*) (*Figure 1B*) that has been tightly linked to initiating escape behaviors (*Fotowat et al., 2011*). This characteristic firing profile is maintained even when an approaching stimulus is embedded in a random motion background, suggesting that the LGMD may be able to effectively segment visual objects (*Silva et al., 2015*; *Yakubowski et al., 2016*). In contrast, the LGMD responds only weakly to a stimulus whose angular size increases linearly in time,

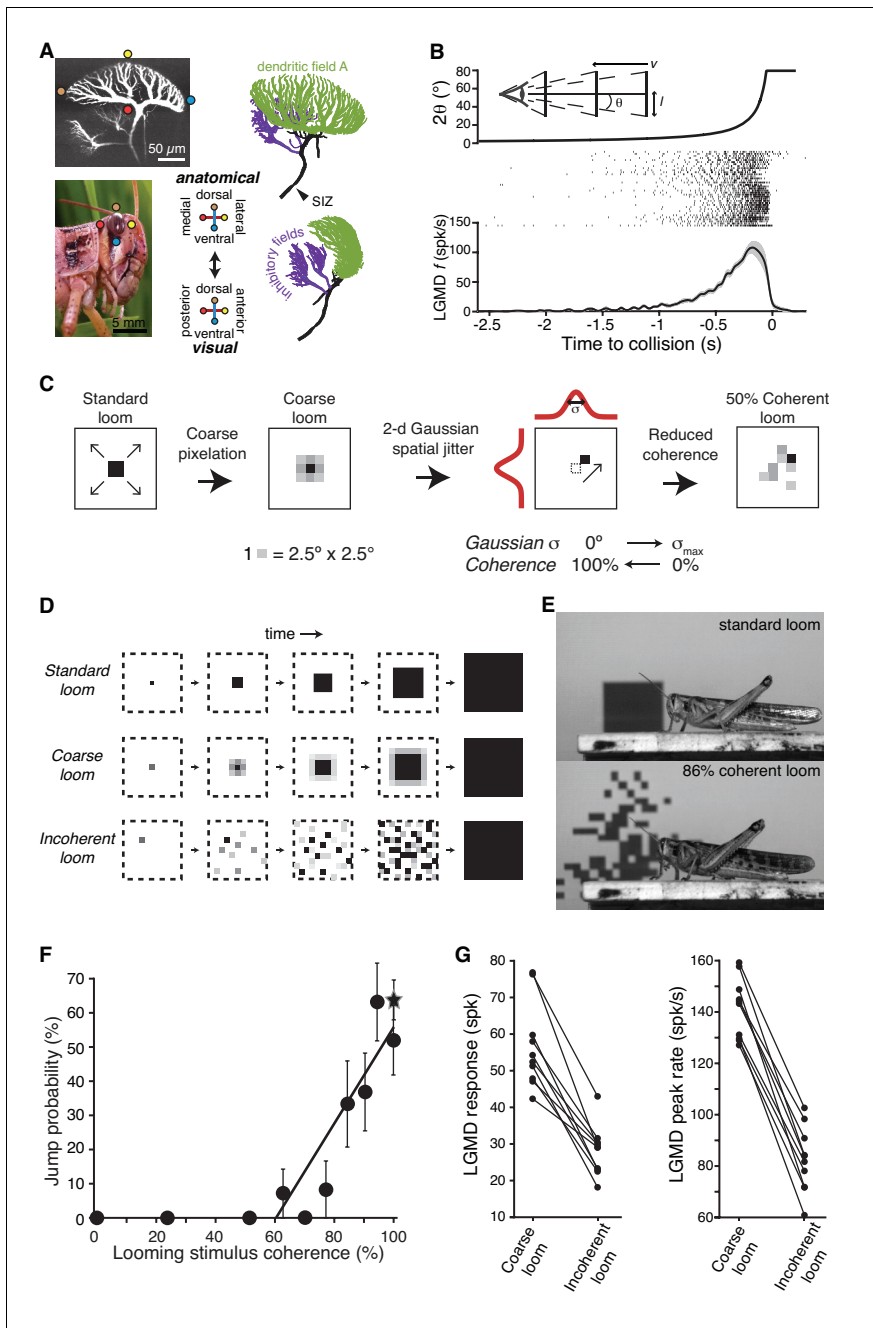

**Figure 1.** LGMD responses and escape behavior are sharply tuned to the spatial coherence of looming stimuli. (A) LGMD 2-photon scan (top, left), eye close-up of *Schistocerca americana* (bottom, left), rostral, and lateral view of a LGMD reconstruction used for modeling (top and bottom right). Excitatory dendritic field in green, SIZ: spike initiation zone. Colored dots illustrate the retinotopic mapping of excitatory inputs to the LGMD. (B) Top, schematic of visual stimulus, half-size *l*, approach speed *v*, half-angular subtense at the eye, *θ*. Note the non-linear increase in angular subtense (*2θ*), characteristic of looming stimuli. Middle, spike rasters of the LGMD responses to looming stimuli. Bottom, mean instantaneous firing rate (*f*) of LGMD looming response. Shaded area is ±1 sem. (C) The coherence of looming stimuli was altered by first applying a coarse pixelation to create photoreceptor sized pixels. Then, a zero-mean random shift was added to the position of these coarse pixels to generate the reduced coherence stimuli. The standard deviation of the random shifting (in degrees) determined the reduction in coherence, with $\sigma_{max} = 40°$ for electrophysiological experiments and $\sigma_{max} = 80°$ for behavioral ones (Materials and methods). (D) Illustration of coherent and incoherent stimuli. For coarse looms (middle row) grayscale levels are set so that luminance in each coarse pixel is equal to that of standard looms in every frame. For reduced coherence
*Figure 1 continued on next page*

*Figure 1 continued*

looms (bottom), the spatial locations of the coarse pixels were altered. (E) Video frames from presentation of standard looming (top) and 86% coherent (bottom) stimuli. (F) Jump probability increased sharply with stimulus coherence above 50% (r = 0.91, p=5.9·10$^{-4}$), 202 trials from 66 animals. Circles are data from coarse and reduced spatial coherence stimuli; star shows response to standard looms. (G) The LGMD's spike count (p=2.5·10$^{-4}$, Wilcoxon rank sum, WRS) and peak firing rate (p=1.9·10$^{-4}$, WRS) were lower for 0% coherent than 100% coherent looming stimuli (N = 10).

DOI: https://doi.org/10.7554/eLife.34238.003

The following source data and source codes are available for figure 1:

**Source code 1.** A Matlab script that will import the data in *Figure 1—source data 1* and generate the plots in *Figure 1*.

DOI: https://doi.org/10.7554/eLife.34238.004

**Source data 1.** An.xlsx spreadsheet with source data plotted in *Figure 1*.

DOI: https://doi.org/10.7554/eLife.34238.005

corresponding to an object decelerating during approach (*Hatsopoulos et al., 1995*; *Simmons and Rind, 1992*).

Synaptic inputs onto the LGMD are physically segregated into three dendritic fields, two of which receive non-retinotopically organized inhibitory inputs (*Strausfeld and Naessel, 1981*). The third one, dendritic field A, receives excitatory inputs originating from each ommatidium (facet) on the ipsilateral compound eye in a precise retinotopic projection (*Krapp and Gabbiani, 2005*; *Peron et al., 2009*; *Zhu and Gabbiani, 2016*) (*Figure 1A*). These excitatory synaptic inputs are segregated by ommatidia and arranged in columnar fashion over an entire visual hemifield, so the LGMD's dendritic arbor has access to the entire spatial visual pattern activated by an approaching stimulus. Like cortical neurons that often receive inputs from tens of thousands of synapses spread across their dendritic arbors, little is known on whether the LGMD detects the spatial patterning of its synaptic input. The precise retinotopy of field A (*Peron et al., 2009*; *Zhu and Gabbiani, 2016*) means that the spatial pattern of synaptic inputs is directly determined by that of the visual stimulus, offering the possibility to experimentally control the synaptic patterning in vivo by changing the spatial aspect of visual stimuli. Thus, we can examine both the LGMD's ability to discriminate spatial patterns consisting of thousands of synaptic inputs and the dynamic membrane properties of its dendrites. Here, we study how the LGMD discriminates the spatial coherence of approaching objects, how hyper-polarization-activated cyclic nucleotide-gated nonselective cation (HCN) channels within the retinotopic dendrites interact with other membrane channels to enhance this discrimination, and how this neural selectivity influences the animal's ability to effectively avoid approaching predators.

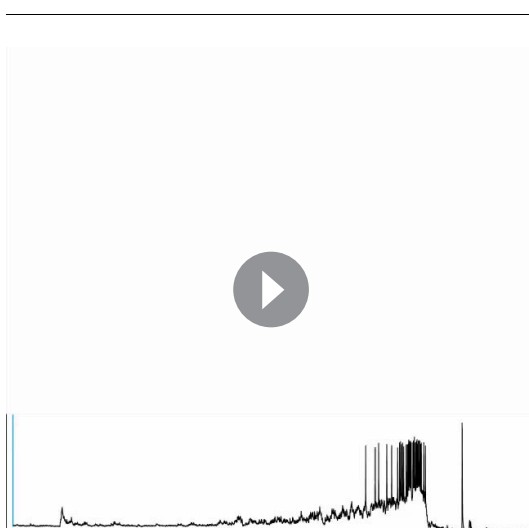

**Video 1.** Looming stimulus with synchronized LGMD membrane potential. At top is a standard looming stimulus with an *l*/|*v*| value of 50 ms. Beneath is the recorded membrane potential of a LGMD neuron during presentation of this stimulus. The vertical blue bar marks the current time of the stimulus. The last burst of activity is caused by the removal of the final black square (not shown in movie). In an experiment, the stimulus appears smoother due to the 200 frame/s refresh rate of the video monitor instead of the 30 frames/s shown here.

DOI: https://doi.org/10.7554/eLife.34238.006

## Results

### Tuning of the LGMD and escape behavior to stimulus coherence

We started with the question of whether the spatial pattern of approaching objects influences

escape behavior. To control the stimulus pattern, as in our earlier work (*Jones and Gabbiani, 2010*), we generated stimuli equivalent to standard looming stimuli but pixelated at the spatial resolution of photoreceptors on the retina, called 'coarse' looming stimuli (*Figure 1C,D*). The LGMD receives similar synaptic excitation and responds equally for standard and coarse looming stimuli (*Jones and Gabbiani, 2010*). We could then alter the coherence of these stimuli with minimal change to the temporal pattern of activation experienced by individual photoreceptors by adding a random spatial jitter to each 'coarse pixel' (*Figure 1C*). Spatial stimulus coherence was varied from random (0%) to perfectly coherent (100% = standard or coarse looming stimulus; see Materials and methods; *Videos 1* and *2*). Looming stimuli with full and reduced coherence were presented to unrestrained animals while recording the probability of escape jumps (*Figure 1E*, *Video 3*). Locusts showed a strong behavioral selectivity to spatial coherence; stimuli with less than 50% spatial coherence elicited no escape jumps, but jump probability increased rapidly with coherence above 50% (*Figure 1F*). The firing rate of the LGMD was also highly sensitive to stimulus coherence, with sharply reduced spike count and peak spike rate at 0% coherence (*Figure 1G*). Thus, the spatial coherence of an approaching object determines both the LGMD's response and the animal's decision of whether to escape.

## HCN channels in dendritic field A are implicated in coherence tuning

None of the known properties of the LGMD or its presynaptic circuitry could explain this spatial selectivity. Previous experiments showed that the strength of excitatory inputs encodes the temporal characteristics of the approaching object by tracking local changes in luminance independent of their spatial pattern (*Jones and Gabbiani, 2010*). Additionally, the spatial clustering of synaptic inputs that occurs with coherent stimuli reduced summation in simulations of passive LGMD dendrites (*Peron et al., 2009*), as further elaborated below. The LGMD's selectivity for the spatial characteristics of an approaching object are therefore likely determined by active processing within the dendrites of field A. No active conductances, however, have yet been characterized within these dendrites. Evidence suggests that neither the fast $Na^+$ nor $Ca^{2+}$ channels that produce supralinear summation in other neurons are present there (*Jones and Gabbiani, 2010*; *Peron and Gabbiani, 2009*). Since in many cells HCN channels influence dendritic computations, and previous experiments suggested putative HCN channels within the LGMD (*Gabbiani and Krapp, 2006*), we hypothesized that HCN channels within field A might be involved in discriminating the spatial coherence of approaching objects. Specifically, HCN channels narrow the membrane's temporal and spatial integration window to excitatory synaptic currents over an extended dendritic harbor. As approaching objects expand toward collision time, the closing of HCN channels could broaden the integration window and thus provide a slow positive feedback mechanism to tune a neuron to the visual stimuli associated with approaching objects.

To test for the presence of HCN channels, we used current and voltage steps, as well as application of known channel blockers and modulators (*Robinson and Siegelbaum, 2003*), during visually guided recordings from each of LGMD's three dendritic fields and near the spike initiation zone (SIZ; *Figure 2A*). Hyperpolarization of field A produced a characteristic rectifying sag, which was abolished by the HCN channel blockers ZD7288 (*Figure 2B*) and $Cs^+$ (*Figure 2—figure supplement 1A,B*). Applying step currents that generated similar peak hyperpolarization (*Figure 2C*, Materials and methods), produced a larger, faster sag in field A than in either field B

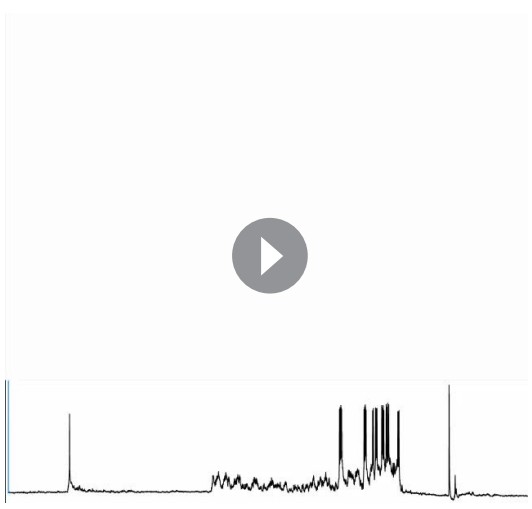

**Video 2.** Spatially incoherent coarse looming stimulus with synchronized LGMD membrane potential. At top is a 0% coherence coarse looming stimulus with an $l/|v|$ value of 50 ms and 2° coarse pixels. Beneath is the recorded membrane potential of a LGMD neuron during presentation of this stimulus. The vertical blue bar marks the current time of the stimulus. The last burst of activity is caused by the removal of the final black square (not shown in movie).
DOI: https://doi.org/10.7554/eLife.34238.007

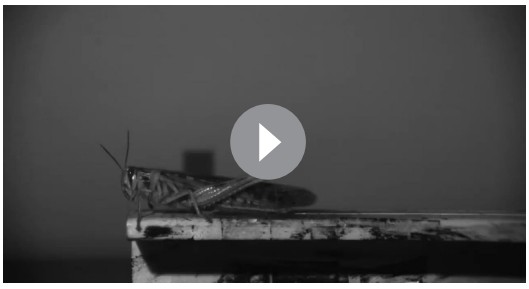

**Video 3.** Escape jump from a looming stimulus. Video of an escape jump from a standard looming stimulus with an $l/|v|$ value of 80 ms. Video was recorded at 200 frames/s and is slowed to 60 frames/s. The animal had received a control saline injection in the optic lobe of the right eye.

DOI: https://doi.org/10.7554/eLife.34238.008

or C, or near the SIZ. To quantify these results across our sample population, we measured sag amplitude from the peak hyperpolarization elicited by these step currents and fitted single exponentials to the sag time course (*Figure 2—figure supplement 1C*). On average, sags recorded from field A without HCN channel block were significantly larger and faster than those recorded from other neuronal regions (*Figure 2D,E*; compare Field A control, vs. trunk and Field B, C). Similarly, block of HCN channels by ZD7288 in field A removed these sags (*Figure 2D,E*; compare Field A control vs. ZD7288). This shows a greater effect of HCN channels' conductance ($g_H$) in field A, consistent with HCN channels being localized there and the current passively propagating to the rest of the LGMD.

Within the dendrites, back-propagating action potentials decay with electrotonic distance from the SIZ. Their amplitude was thus used as a measure of electrotonic distance from the SIZ to the recording location. Recordings at different locations within the dendritic trunk and field A revealed an increase in sag with electrotonic distance from the SIZ (*Figure 2F*) suggesting a higher channel density distally in field A. To characterize the channel kinetics, field A dendrites were voltage clamped. As HCN channels are distinctive in their activation range and time course we were able to measure and fit their currents (see Materials and methods; *Figure 2—figure supplement 1D*), revealing an activation curve (*Figure 2G*) and time constant (*Figure 2H*) similar to that of HCN2 channels (*Robinson and Siegelbaum, 2003*). Next, we tested modulation of the HCN channels by cAMP. Application of cAMP shifted the half-activation potential ($v_{1/2}$) of $g_H$ from $-77.6 \pm 3.8$ to $-73.4 \pm 2.2$ mV (mean ±sd; *Figure 2G*) and slightly increased activation at rest (*Figure 2I*). Both changes, however, were not different from controls (p=0.18 and p=0.076, respectively). These observations are in agreement with a recent report of a decrease in exogenous cAMP modulation of HCN channels in vivo post-developmentally, likely due to saturation of naturally occurring cAMP levels (*Khurana et al., 2012*). In contrast, ZD7288 application unambiguously abolished resting $g_H$ activation (*Figure 2I*).

To examine whether these HCN channels could be responsible for spatial discrimination, we presented visual stimuli before and after their pharmacological blockade. For standard looming stimuli, $g_H$ was excitatory with responses reduced by 61% after HCN blockade (*Figure 2J*). Responses to localized luminance transients, however, were similar before and after blockade of $g_H$ (*Figure 2—figure supplement 2A–C*). Next, we quantified the responses to looming stimuli of varying coherence in control and after HCN blockade by computing the spike counts elicited over each entire trial. Since the LGMD did not exhibit any significant spontaneous activity, changes in spike counts were entirely caused by changes in the stimulus coherence. Under control conditions, there was a large increase in LGMD response with stimulus coherence which was reduced after ZD7288 blockade (42% for coarse looming stimuli; *Figure 2K*). For each experiment, we defined coherence preference as the slope of the linear fit to the number of spikes fired by the LGMD as a function of coarse looming stimulus coherence. For every animal tested, the coherence preference was reduced after $g_H$ blockade, decreasing from a median of 0.24 to 0.06 spikes per percent coherence (*Figure 2L*; similar results were observed for the peak firing rate, see *Figure 2—figure supplement 2D,E*). To compare this relationship across animals, we normalized responses to the control response to fully coherent coarse stimuli before averaging across animals (*Figure 2M*). LGMD responses consistently increased less with stimulus coherence after $g_H$ block (*Figure 2M*). After blockade, the mean response to all stimuli fell within ±1 sd of the mean control response to 0% coherence (p=0.10; KW). As explained below, this change in selectivity was reproduced by a biophysical model of the LGMD (*Figure 2M*, dashed lines). To ensure that these effects were intrinsic to the LGMD, we ascertained that blocking $g_H$ with intracellular $Cs^+$ application also reduced the coherence selectivity (*Figure 2—figure supplement 2F*; Materials and methods).

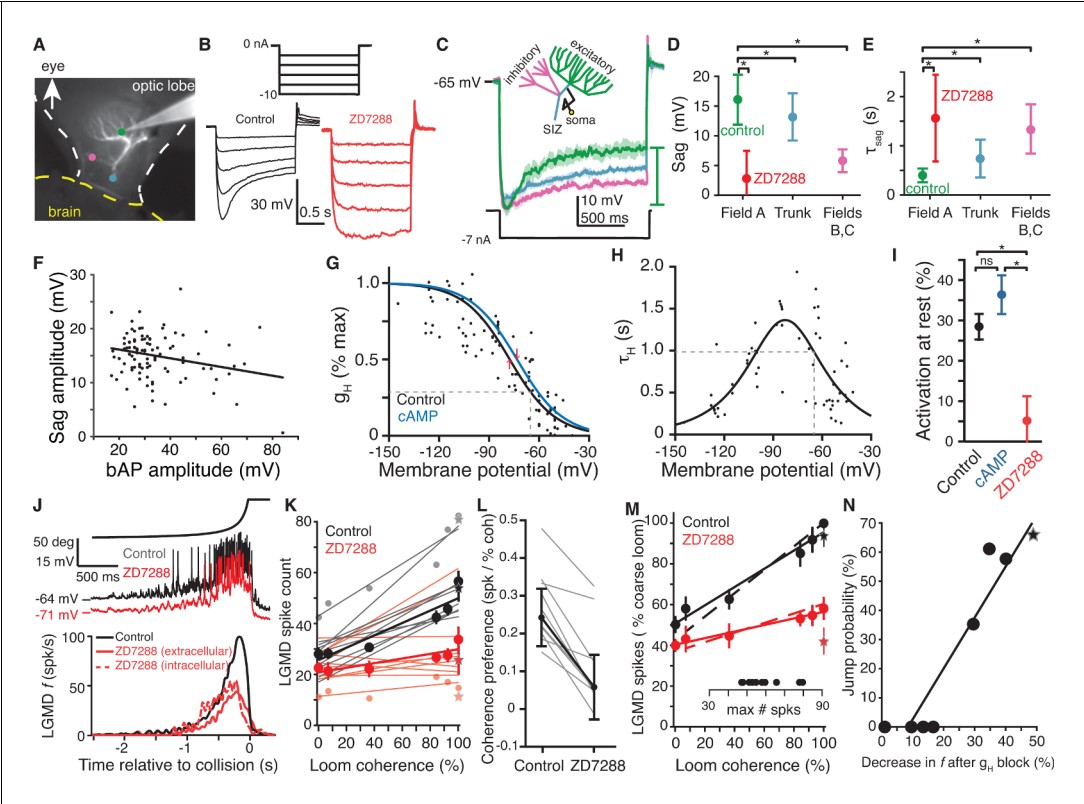

**Figure 2.** HCN channels in dendritic field A are responsible for spatial coherence sensitivity. (A) Image showing the LGMD stained in vivo with Alexa 594 and the recording electrode with tip at the green dot. Colored dots indicate the recording locations of traces shown in (C). (B) Hyperpolarizing current steps (top) injected in field A generate a characteristic rectifying sag in control recordings (left), but application of ZD7288 completely removed the sag (right). (C) Schematic of the LGMD's dendritic subfields and example traces showing larger rectifying sag in field A than in field C or near the SIZ. Solid lines are the average response with shaded region of ±1 sd. Sag amplitude was measured as the amount of rectification from peak hyperpolarization to steady state, as indicated by the green bar. (D) Sag amplitudes following steps from rest yielding peak hyperpolarizations between −95 and −115 mV are consistently larger in field A (N = 82,58) than after ZD7288 application (N = 13,9) or in recordings from the trunk (N = 11,13) or inhibitory subfields (N = 6,6; *: p<0.05, KW-MC). (E) The sag time constant for these responses was smaller in field A (N = 82,58) than after ZD7288 application (N = 13,9) or in recordings from the trunk (N = 11,13) or inhibitory subfields (N = 6,6; *: p<0.01, KW-MC). For (D, E), points are median and error bars are one mad. (F) Sag amplitude along the trunk and in field A decreased with increased backpropagating action potential (bAP) amplitude, a measure of electrotonic distance from the spike initiation zone (r = −0.25, p=0.01, N = 104,69). (G) Activation curve of $g_H$ measured in voltage clamp. Black line is control (N = 8,7; $v_{1/2}$ = −77.6 mV, 28% of max at RMP, dashed line; $R^2$ = 0.69) and blue line after local application of cAMP (N = 6,6; $v_{1/2}$ = −73.4 mV, 35% of max at RMP). Red arrows indicate shift in $v_{1/2}$. (H) Time constant of $g_H$ from voltage clamp recordings (N = 8,7; $\tau_H$max = 1.34 s, at −83 mV; $\tau_H$ = 985 ms at RMP, dashed line; steepness = 20 mV; $R^2$ = 0.61). (I) Resting activation of HCN channels, relative to max, displayed as mean and sem (control N = 82,58; cAMP N = 6,6; ZD7288 N = 13,10; *: p<0.001, ns: p=0.076, unpaired t-test). (J) Intracellular recordings of LGMD's membrane potential in response to looming stimuli show decreased RMP and activation after blockade of $g_H$ (top). Bottom, mean instantaneous firing rates (f) in response to looming stimuli declined after intra- or extra-cellular application of ZD7288 (N = 10,10 p=4.1·10⁻⁵, WRS). (K) Each line shows the linear fit to the LGMD response of an animal before (black) and after (red) puffing ZD7288 (N = 10). Half-tone dots show representative data from one animal for the corresponding fit line. Thick lines and dots are population averages. Stars are standard loom. The slope of control data was higher than after $g_H$ block (p=0.001), but the intercepts were not different (p=0.18; ANCOVA, N = 10,10). (L) For all experiments coherence preference decreased after $g_H$ blockade (N = 10, gray lines). Coherence preference was calculated as the increase in spike count per percent increase in stimulus coherence. Black dots and lines show the average coherence preference decreased by 0.18 spikes per percent stimulus coherence (p=7.9·10⁻⁵, paired t-test). (M) After $g_H$ block the slope of coherence-dependent increase was reduced from 0.45 to 0.16 (p=3.7·10⁻⁴, ANCOVA test of slopes, N = 10,10). Solid lines and dots are coarse loom data, stars are standard loom, error bars are ±1 sem, and dashed lines are compartmental simulation results. Insets show plot normalization values. (N) Jump probability for a stimulus correlates strongly with its $g_H$-dependent increase in firing (r = 0.94, p=4.1·10⁻⁴). Circles are data from coarse and reduced spatial coherence stimuli; star shows response to standard looms. N: number of recordings, number of animals.
DOI: https://doi.org/10.7554/eLife.34238.009

The following source data, source code and figure supplements are available for figure 2:

**Source code 1.** A Matlab script that will import the data in *Figure 2—source data 1* and generate the plots in *Figure 2*.
DOI: https://doi.org/10.7554/eLife.34238.012

**Source code 2.** A Matlab script that will import the data in *Figure 2—source data 2* and generate the plots in *Figure 2—figure supplement 1*.

*Figure 2 continued on next page*

*Figure 2 continued*

DOI: https://doi.org/10.7554/eLife.34238.013

**Source code 3.** A Matlab script that will import the data in *Figure 2—source data 3* and generate the plots in *Figure 2—figure supplement 2*.

DOI: https://doi.org/10.7554/eLife.34238.014

**Source data 1.** An.xlsx spreadsheet with data plotted in *Figure 2*.

DOI: https://doi.org/10.7554/eLife.34238.015

**Source data 2.** An.xlsx spreadsheet with source data plotted in *Figure 2—figure supplement 1*.

DOI: https://doi.org/10.7554/eLife.34238.016

**Source data 3.** An.xlsx spreadsheet with source data plotted in *Figure 2—figure supplement 2*.

DOI: https://doi.org/10.7554/eLife.34238.017

**Figure supplement 1.** ZD7288 and $Cs^+$ block $g_H$ within the LGMD.

DOI: https://doi.org/10.7554/eLife.34238.010

**Figure supplement 2.** Effects of $g_H$ on local or looming stimuli assessed with block by ZD7288 or $Cs^+$.

DOI: https://doi.org/10.7554/eLife.34238.011

Next, we compared the jump probabilities at each coherence level (*Figure 1F*) with the $g_H$-dependent increase in firing for that coherence level (difference between control and HCN block in *Figure 2M*). This revealed a strong correlation between physiology and behavior (*Figure 2N*). Furthermore, responses to faster looming stimuli, which fail to produce escape behaviors before the projected time of collision (*Fotowat and Gabbiani, 2007*), showed a smaller $g_H$-dependent increase in firing (*Figure 2—figure supplement 2G*). Therefore, $g_H$ increased responses specifically to stimuli which evoke escape, suggesting that the $g_H$-dependent enhancement produced the escape selectivity.

## HCN channels mediate coherence tuning of escape behaviors

Having found that $g_H$-dependent increase in firing is strongly correlated with jump probabilities (*Figure 2N*), we sought a direct test of the hypothesis that $g_H$ within the LGMD played a role in the animals' escape from approaching objects. So, we blocked $g_H$ in the LGMD in freely behaving animals (Materials and methods). As a control, we developed a chronic recording technique allowing us to monitor the descending LGMD output during escape behaviors before and after $g_H$ blockade.

Blocking $g_H$ in the LGMD reduced escape behavior by 53% for standard looming stimuli compared to saline injection (*Figure 3A*, left two dots). The coherence preference was also removed by blockade of $g_H$: standard looming stimuli no longer produced a higher percentage of escape than reduced coherence stimuli (*Figure 3A*, red dots). That these behavioral changes were caused by $g_H$ blockade within the LGMD was further confirmed by examination of the LGMD's firing pattern. $g_H$ blockade by ZD7288 decreased responses to both standard looming stimuli and 86% coherent stimuli (*Figure 3B,C*). The reduction in firing in the freely moving animals was less than that in the restrained preparation (36% and 60%, respectively), which might be due to an incomplete block of $g_H$ after stereotactic injection compared to visually guided puffing (see Materials and methods) or differences in arousal state. To test this, we used the stereotactic injection procedure in restrained animals and saw a 56% reduction in looming responses (*Figure 3D*) suggesting the difference in firing rate change was more likely due to a difference in behavioral state. Our ability to produce a change in behavior of freely moving animals from blockade of $g_H$ was confirmed by simultaneous extracellular recordings revealing a LGMD firing rate change resembling that of intracellular drug application, verification that the surgical procedures did not reduce the response, and postmortem anatomical verification that drug application occurred within the region encompassing the LGMD's dendrites (*Figure 3—figure supplement 1*).

## HCN channels affect membrane properties and synaptic summation

The precise biophysical mechanisms by which HCN channels could impart the selective enhancement of coherent stimulus responses are not immediately obvious. HCN channels are not known to increase summation of spatially coherent inputs, and often $g_H$ has net inhibitory effects (*Robinson and Siegelbaum, 2003*; *Poolos et al., 2002*). To determine how HCN channels produced the selective enhancement of looming responses, we investigated the effects of $g_H$ on membrane excitability within field A. $g_H$ increased the resting membrane potential (RMP) by ~6 mV in field A,

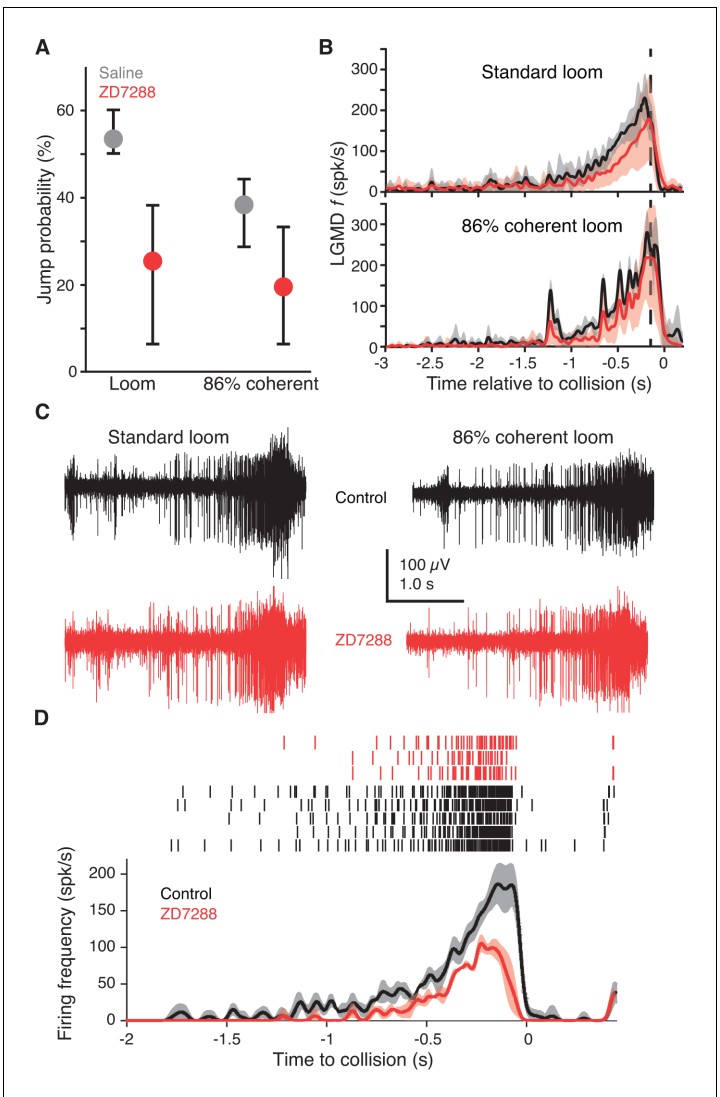

**Figure 3.** Blocking HCN channels removed coherence preference of escape behavior. (**A**) Jump probability for coherent looming stimuli decreased after injecting ZD7288 into the lobula, compared to saline injection (p=0.008, ASL). Error bars are bootstrapped 95% confidence intervals. Responses to 86% coherent stimuli after ZD7288 injection were not significantly different from responses after saline injection (p=0.08, ASL) or standard looming responses after ZD7288 injection (p=0.34, ASL). Saline injection: 48 trials from five animals, ZD7288 injection: 41 trials from five animals. (**B**) LGMD instantaneous firing rates (f) during jump experiments decreased after ZD7288 injection. ZD7288 decreased responses to both stimuli (p=0.019 for loom, p=0.015 for 86% coherent, WRS; N = 3). Vertical dashed lines show the average time of jump. (**C**) Example extracellular recordings during jump experiments before and after ZD7288 application. The control responses displayed have 182 and 165 spikes, and responses after ZD7288 have 128 and 122 spikes for the standard and 86% coherent loom, respectively. (**D**) Rasters and instantaneous firing rates after stereotactic injection of ZD7288 through the eye in restrained animals; LGMD responses were reduced similarly to intracellular and visually guided application (*Figure 2J*), demonstrating that the stereotactic injection method successfully targeted the LGMD.

DOI: https://doi.org/10.7554/eLife.34238.018

The following source data, source code and figure supplements are available for figure 3:

**Source code 1.** A Matlab script that will import the data in *Figure 3—source data 1* and generate the plots in *Figure 3*.

DOI: https://doi.org/10.7554/eLife.34238.020

**Source code 2.** A Matlab script that will import the data in *Figure 3—source data 2* and generate the plots in *Figure 3—figure supplement 1*.

DOI: https://doi.org/10.7554/eLife.34238.021

*Figure 3 continued*

**Source data 1.** An.xlsx spreadsheet with data plotted in *Figure 3*.
DOI: https://doi.org/10.7554/eLife.34238.022
**Source data 2.** An.xlsx spreadsheet with source data plotted in *Figure 3—figure supplement 1*.
DOI: https://doi.org/10.7554/eLife.34238.023
**Figure supplement 1.** Change in escape behavior due to $g_H$ blockade within the LGMD.
DOI: https://doi.org/10.7554/eLife.34238.019

which would bring the neuron closer to spike threshold (*Figure 4A*). Blockade also revealed $g_H$ to decrease input resistance by 50% and the membrane time constant ($\tau_m$) by 30% (*Figure 4B,C*), which should substantially reduce the temporal summation of excitatory postsynaptic potentials (EPSPs), as occurs in cortical pyramidal neurons (*Magee, 1998*; *Mishra and Narayanan, 2015*). To confirm this point, we injected currents yielding membrane depolarizations with the same time course as EPSPs to generate 'simulated EPSPs' (sEPSPs; *Figure 4D*). After $g_H$ blockade, summation from the first to fifth sEPSP increased for all tested delays (*Figure 4E*; the dashed lines are from the biophysical model described below). Additionally, the integrated sEPSPs normalized by the integrated current increased by 77% (*Figure 4F*). This normalized integral generates a measure of effective input resistance for the sEPSPs which was similar to the input resistance to step currents, but with a slightly larger increase after HCN blockade (compare *Figure 4B and F*). Neither before nor after $g_H$ blockade was supralinear summation ever seen in LGMD dendrites. Thus, the mix of local excitatory and inhibitory electrotonic effects of $g_H$ does not provide any simple explanation for the large enhancement in looming responses or the conveyed coherence selectivity.

## K$^+$channels complement HCN channels in generating coherence tuning

It may seem counterintuitive that $g_H$ increased looming responses twofold despite decreasing sEPSP amplitude and temporal summation by half. To explain this apparent contradiction, we considered interactions between HCN and other dendritic channels. In several systems, HCN channels have indirect excitatory effects through inactivation of co-localized voltage-gated K$^+$ channels (*Mishra and Narayanan, 2015*; *Khurana et al., 2011*; *MacLean et al., 2005*; *Amendola et al., 2012*). To test whether this was also the case in dendritic field A of the LGMD, we measured visual responses in the presence of 4-aminopyridine (4AP), a blocker of inactivating K$^+$ channels (*Storm, 1988*). Application of 4AP, either intracellularly or extracellularly, increased the resting membrane potential in field A by 2–5 mV and the spiking response and instantaneous firing rate to standard looming stimuli (*Figure 5A*). Application of 4AP also increased responses to coarse looming stimuli of varying degree of coherence, but responses to fully coherent looming stimuli increased the least (*Figure 5B*). A similar result was observed after normalizing responses to the control response to fully coherent coarse stimuli before averaging across animals (*Figure 5C*). This relative increase in incoherent responses after blocking inactivating K$^+$ channels was also reproduced in a biophysical model (*Figure 5C*, dashed lines; see below). The complementary effects of HCN and K$^+$ channels was best revealed by plotting their relative changes to looming responses, shown as the percent difference from block to control (*Figure 5D*). Thus, while HCN channels predominantly boosted responses to coherent stimuli, inactivating K$^+$ channels mainly decreased responses to incoherent ones.

The increase in RMP caused by $g_H$ (*Figure 4A*) could result in a change in the resting inactivation level of the 4AP sensitive K$^+$ channels. To test whether the effects of $g_H$ on coherent stimuli were primarily due to the shift in the RMP, we hyperpolarized the LGMD during visual stimuli to a potential like that achieved by HCN blockade (~6 mV, see above). However, lowering the RMP without the changes to input resistance and membrane time constant caused by $g_H$ blockade (*Figure 4B,C*) only produced a modest reduction in coherence preference, with responses to standard looming stimuli reduced by 20% vs. 61% after ZD7288 blockade (p=0.03, WRS; see above and *Figure 5—figure supplement 1*). This reduction in coherence preference was less than that produced by blockade of either dendritic channel and was independent of stimulus coherence (p=0.36, KW; *Figure 5D*). This result corroborates the idea that dynamic changes of the $g_H$ conductance occurring during looming stimulation and their effects on electrical compactness and membrane time constant play a central role in coherence selectivity.

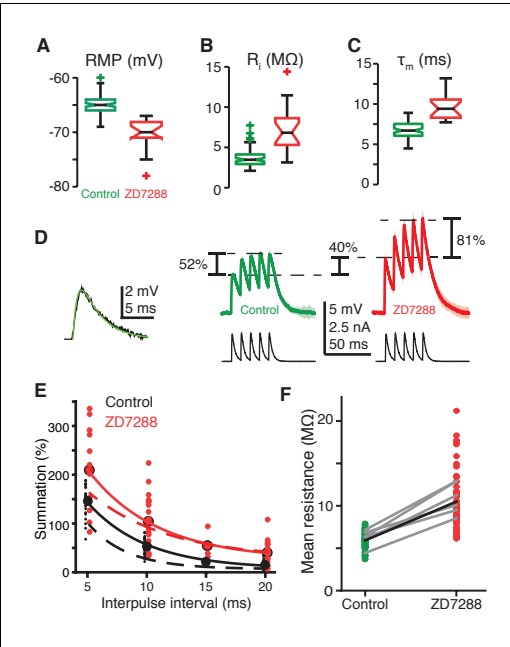

**Figure 4.** $g_H$ conductance decreases EPSP amplitudes and summation. (**A**) RMP in field A decreased after blockade of $g_H$ by ZD7288 (control N = 82,58; ZD7288 N = 15,9; p=1.9·10$^{-9}$, WRS). (**B**) Input resistance ($R_i$) in field A increased after $g_H$ blockade (control N = 78,58; ZD7288 N = 17,10; p=6.1·10$^{-7}$, WRS). (**C**) Membrane time constant ($\tau_m$) also increased in field A after $g_H$ block (control N = 82,58; ZD7288 N = 16,10; p=1.3·10$^{-8}$, WRS). (**D**) Left, example visual EPSP (black) and sEPSP (green). Right, example responses to a series of 5 sEPSPs with 10 ms interpulse interval. On average ZD7288 block of $g_H$ led to 51% increase in 1st sEPSP amplitude (p=6.0·10$^{-4}$, WRS) and subsequent summation of sEPSPs from 53% in control to 104% (p=0.001, WRS; control N = 14,7; ZD7288 N = 9,5). (**E**) Summation increased after $g_H$ removal for all interpulse intervals. Points and solid lines show experimental data, dashed lines are simulation results. (**F**) The integrated membrane potential ($V_m$) normalized by the integrated current (total charge) increased an average of 77% after ZD7288 (thick black line; p=1.7·10$^{-19}$, WRS, N = 14,7 in control and 9,5 in ZD7288). Gray lines are from six recordings held through puffing (p≤0.01, paired t-test).

DOI: https://doi.org/10.7554/eLife.34238.024

The following source data and source codes are available for figure 4:

**Source code 1.** A Matlab script that will import the data in **Figure 4—source data 1** and generate the plots in **Figure 4**.

DOI: https://doi.org/10.7554/eLife.34238.025

**Source data 1.** An.xlsx spreadsheet with data plotted in **Figure 4**.

DOI: https://doi.org/10.7554/eLife.34238.026

To further confirm that the inactivating K$^+$ channels were exerting a spatially dependent effect on synaptic integration, we measured subthreshold activity in dendritic field A before and after 4AP application while presenting looming stimuli with varying degrees of coherence. As illustrated in **Figure 6A** (top), we measured during a given period of the visual stimulus (vertical green lines) the average membrane depolarization before and after application of 4AP (grey horizontal lines). During this period, a distinct group of coarse pixels were decreasing in luminance. As illustrated in the two bottom panels of **Figure 6A**, we measured the mean angular distance of each currently changing pixel from the nearest previously darkened pixel (red lines). A short distance example is depicted on the left and large one on the right of **Figure 6A**. In the control condition (the top panels), the membrane potential was closer to the baseline for the larger angular distance (compare left and right black traces). This decrease in depolarization was attenuated after 4AP application (blue traces). Since mean angular distance increased on average with decreasing coherence, we could obtain a broad sample of distances by carrying out this analysis across trials and animals. This revealed that the membrane depolarization systematically decreased with increasing stimulus distance in the control condition, but this effect was abolished after 4AP (**Figure 6B**). We repeated this process for a total of six distinct time periods for each looming stimulus. Throughout stimulus expansion, the more dispersed excitatory inputs were, the less dendritic depolarization they produced in control conditions, a feature absent after 4AP application (**Figure 6—figure supplement 1**). These results are summarized across the six time periods in **Figure 6C**, by normalizing angular distance and membrane potential depolarization since their ranges vary over the stimulus time course. We further quantified the change in depolarization caused by currently changing coarse pixels as a function of their mean angular distance to fully darkened ones by computing the slope of the linear fits between these two quantities (**Figure 6D**). Smaller slopes were observed in the earlier time windows when angular distances were larger and only a few coarse pixels were changing and, vice-versa, larger slopes were observed in later time windows when distances were smaller, but more pixels were changing their luminance. For each time window, 4AP increased the depolarization produced in field A by more distant stimuli with an average slope difference of 1.05 mV per degree of visual

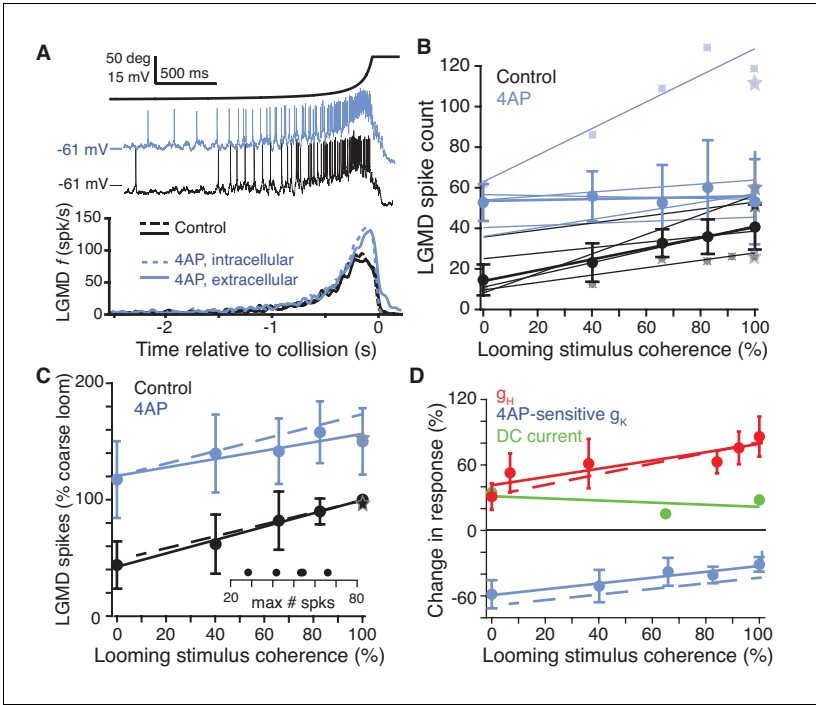

**Figure 5.** A 4AP-sensitive K$^+$ conductance decreases responses to incoherent stimuli. (**A**) The time course of the looming stimulus is indicated on top by its subtended visual angle (2θ, see **Figure 1B**) and (middle) example recordings of LGMD membrane potential in response to looming stimuli before and after local puff of 4AP. Below, average firing rate of the LGMD before and after application of 4AP. Both intracellular and extracellular application had the same effect on looming responses (N = 5,5 for extracellular, N = 3,3 for intracellular). (**B**) Individual linear fits (control: r = 0.93 ± 0.04; 4AP r = 0.64 ± 0.48, mean ±sd) to the LGMD responses before and after puffing 4AP show an increase in firing for all animals, and a decrease in coherence preference for all but one animal. Data points for the top and bottom fit are shown for example. Thicker lines show linear fits to the median response (error bars are ± mad). (**C**) 4AP increased average responses to all stimuli (p=0.002; WRS), and reduced the coherence-dependent increase in firing from 0.57 to 0.36 (p=0.014, ANCOVA test of slopes; N = 5,5). Plotted as in **Figure 2M**. Dashed lines in C) and D) are simulation data. (**D**) To estimate the influence of HCN and 4AP-sensitive K$^+$ channels on firing we calculated the percent change after channel blockade relative to control (channels present). $g_H$ increased responses, with larger increase for coherent stimuli. The K$^+$ channels decreased responses with larger decreases for incoherent stimuli. As comparison, tonic −2.5 nA current injection resulted in smaller effects, with percent change independent of stimulus coherence (p=0.36, KW).

DOI: https://doi.org/10.7554/eLife.34238.027

The following source data, source code and figure supplements are available for figure 5:

**Source code 1.** A Matlab script that will import the data in **Figure 5—source data 1** and generate the plots in **Figure 5**.
DOI: https://doi.org/10.7554/eLife.34238.029

**Source code 2.** A Matlab script that will import the data in **Figure 5—source data 2** and generate the plots in **Figure 5—figure supplement 1**.
DOI: https://doi.org/10.7554/eLife.34238.030

**Source data 1.** An.xlsx spreadsheet with data plotted in **Figure 5**.
DOI: https://doi.org/10.7554/eLife.34238.031

**Source data 2.** An.xlsx spreadsheet with source data plotted in **Figure 5—figure supplement 1**.
DOI: https://doi.org/10.7554/eLife.34238.032

**Figure supplement 1.** Effect of hyperpolarizing current injection on coherence preference.
DOI: https://doi.org/10.7554/eLife.34238.028

separation (**Figure 6E**). These experiments confirm that inactivating K$^+$ channels selectively reduce excitation for the spatially dispersed inputs generated in dendritic field A by incoherent looming stimuli and thus contribute to the selectivity of the neuron to coherently expanding looming stimuli.

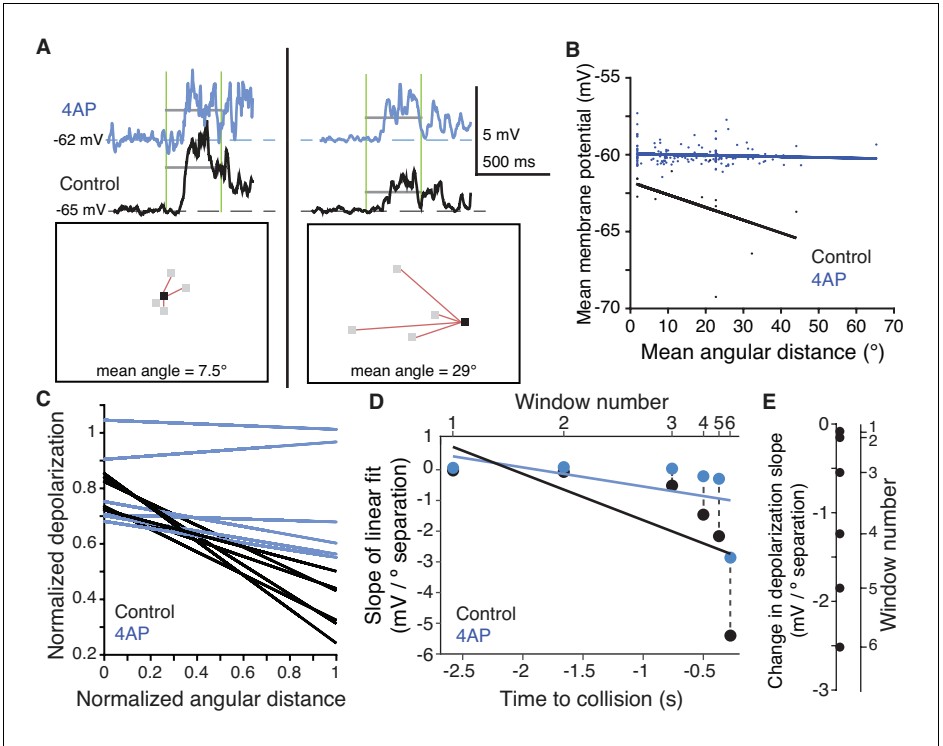

**Figure 6.** A 4AP-sensitive K$^+$ conductance reduces dendritic depolarization for spatially incoherent inputs. (**A**) Example of the first of six time periods in which the mean membrane potential and mean angular distance were measured. Top, traces of the dendritic membrane potential recorded within field A. At left are responses to higher spatial coherence stimuli, at right are responses to lower spatial coherence stimuli. Black traces are control and blue traces are after 4AP application. Vertical green bars mark the time period measured; horizontal grey bars show the mean V$_m$ during the time period. Bottom, example stimulus frames taken within the first time period. Red lines are drawn between newly changing coarse pixels (grey) and the only previously darkened one (black). (**B**) Under control conditions, an inverse relationship is seen, with more spatially dispersed stimuli generating less membrane depolarization. Application of 4AP removed this effect (p=0.0002, ANCOVA test of slopes). (**C**) Membrane potential changes showed a strong negative correlation with distance between stimulated regions in control (r = −0.54, p=4.1·10$^{-8}$), but 4AP application significantly reduced this effect (p=0.006, ANCOVA test of slopes, N = 7,7). Each line pair is from a different time period of the stimulus. (**D**) The reduction in depolarization of distant inputs is shown with a point from each time window. Although the range of distances decreases as the stimulus nears collision (p=1.2·10$^{-6}$, t-test), the reduction in depolarization per degree of separation increases (p=1.7·10$^{-10}$, KW). (**E**) Within each time period, the 4AP-sensitive current caused a decrease in response to spatially distant inputs (p=0.03, signed rank, N = 7,7).

DOI: https://doi.org/10.7554/eLife.34238.033

The following source data, source code and figure supplements are available for figure 6:

**Source code 1.** A Matlab script that will import the data in *Figure 6—source data 1* and generate the plots in *Figure 6*.

DOI: https://doi.org/10.7554/eLife.34238.035

**Source code 2.** A Matlab script that will import the data in *Figure 6—source data 2* and generate the plots in *Figure 6—figure supplement 1*.

DOI: https://doi.org/10.7554/eLife.34238.036

**Source data 1.** An.xlsx spreadsheet with data plotted in *Figure 6*.

DOI: https://doi.org/10.7554/eLife.34238.037

**Source data 2.** Acsv file with source data plotted in *Figure 6—figure supplement 1*.

DOI: https://doi.org/10.7554/eLife.34238.038

**Figure supplement 1.** Relationship between mean stimulus angular distance and membrane potential before and after application of 4AP.

DOI: https://doi.org/10.7554/eLife.34238.034

# Compartmental modeling highlights role of K$^+$ and Ca$^{2+}$channel inactivation in coherence tuning

Detailed biophysical modeling was employed to further understand the biophysical mechanisms by which HCN and inactivating K$^+$ channels allow the LGMD to discriminate spatiotemporal input patterns based on coherence. First, we confirmed that a model of the LGMD with passive dendrites generated a smaller response to retinotopically arranged looming inputs than the same inputs impinging on random dendritic locations both in terms of the mean membrane potential and the instantaneous firing rate (*Figure 7A*, top and bottom panels, respectively). This illustrates why implementing coherence preference is nontrivial: the spatially distributed excitatory inputs that occur during incoherent looming stimuli produce less reduction in driving force, thus generating a larger current from the same synaptic conductance. Adding HCN channels to the dendrites of this model while adjusting the leak conductance to maintain RMP and R$_i$, also resulted in stronger responses to spatially scrambled inputs (*Figure 7B*). As suggested by the results of *Figures 5* and *6*, the subsequent addition of inactivating K$^+$ channels in dendritic field A reduced responses to the spatially scrambled inputs, bringing the model in broad agreement with experimental findings (*Figure 7C*).

More precisely, the model reproduced key experimental results, including the LGMD's preference for spatially coherent inputs and the reduction of this preference after block of g$_H$ (*Figure 2M*; *Figure 5D*); the electrotonic and summation effects of g$_H$ (*Figure 4E*; *Figure 7—figure supplement 1A–C*); the coherence-dependent increase in firing caused by blocking the inactivating K$^+$ channels and their role in suppressing responses to incoherent stimuli (*Figure 5C,D*). In the model, the inactivating K$^+$ channel activity was similar to the K$_D$ current that has been hypothesized to influence dendritic integration in pyramidal and Purkinje neurons (*Storm, 1988*; *Hounsgaard and Midtgaard, 1989*; *Zagha et al., 2010*). This similarity extended to its activity at rest, its influence on subthreshold integration within field A dendrites, its apparent slow inactivation, and its 4AP sensitivity. We thus call it K$_{D-like}$.

During coherent looming stimuli, inputs continue to impinge on nearby dendritic segments for a prolonged period, spreading slowly (*Figure 7D*, top). With spatially incoherent stimuli, inputs spread out over a much larger region of the dendritic arbor (*Figure 7D*, bottom). The dendritic branches receiving the concentrated inputs of a coherent loom depolarize more than the surrounding branches, while the spatially dispersed inputs of an incoherent loom produce a similar level of depolarization across the dendrites (*Figure 7E*).

The prolonged depolarization generated by a coherent loom causes HCN channels to close (*Figure 7F*, dashed black line) and the K$_{D-like}$ channels to inactivate (*Figure 7F*, solid black line). The deactivation of HCN channels leads to increased spatial compactness and summation (*Figure 4*) providing a slow positive feedback while the faster negative feedback provided by K$_{D-like}$ decreases as it inactivates. During spatially incoherent stimuli, however, HCN channels close less and the K$_{D-like}$ channels across the arbor undergo less inactivation (*Figure 7F*, gray lines). In control conditions, K$_{D-like}$ inactivation is due to the resting depolarization from g$_H$ and activity-induced depolarization. After HCN blockade, the lower resting membrane potential reduces the baseline K$_{D-like}$ channel inactivation (*Figure 7F*, red line) so that even with spatially coherent inputs the channels never reach the same level of inactivation. The K$_{D-like}$ channel activation was highest for control looming stimuli and lowest for incoherent looming stimuli (*Figure 7—figure supplement 1D*) as it tracked the membrane potential (*Figure 7E*). However, the overall conductance of the K$_{D-like}$ channels was lowest for coherent looming stimuli contributing to the higher response (*Figure 7—figure supplement 1E*).

Examination of the membrane currents generated by the channels reveals even larger differences. Toward the end of a looming stimulus, the dendrites approach the HCN channel reversal potential, and the net HCN current approaches zero (*Figure 7G*, top). Conversely, the K$^+$ driving force increases during the stimulus approach. As a result, the K$_{D-like}$ channels that remain activatable produce a larger current (*Figure 7G*, bottom). For the coherent stimulus, this late depolarization occurs in the same dendritic region activated by the earlier inputs and since the nearby K$_{D-like}$ channels have already inactivated, it yields little increase in K$^+$ current, irrespective of g$_H$ block. The incoherent inputs, however, impinge onto branches where the channels have not already inactivated, yielding a much larger current.

In addition to these dendritic channels, the model also included low-threshold Ca$^{2+}$ channels (Ca$_T$) and Ca$^{2+}$-dependent K$^+$ channels (K$_{Ca}$) near the SIZ that allowed the LGMD model to fire in

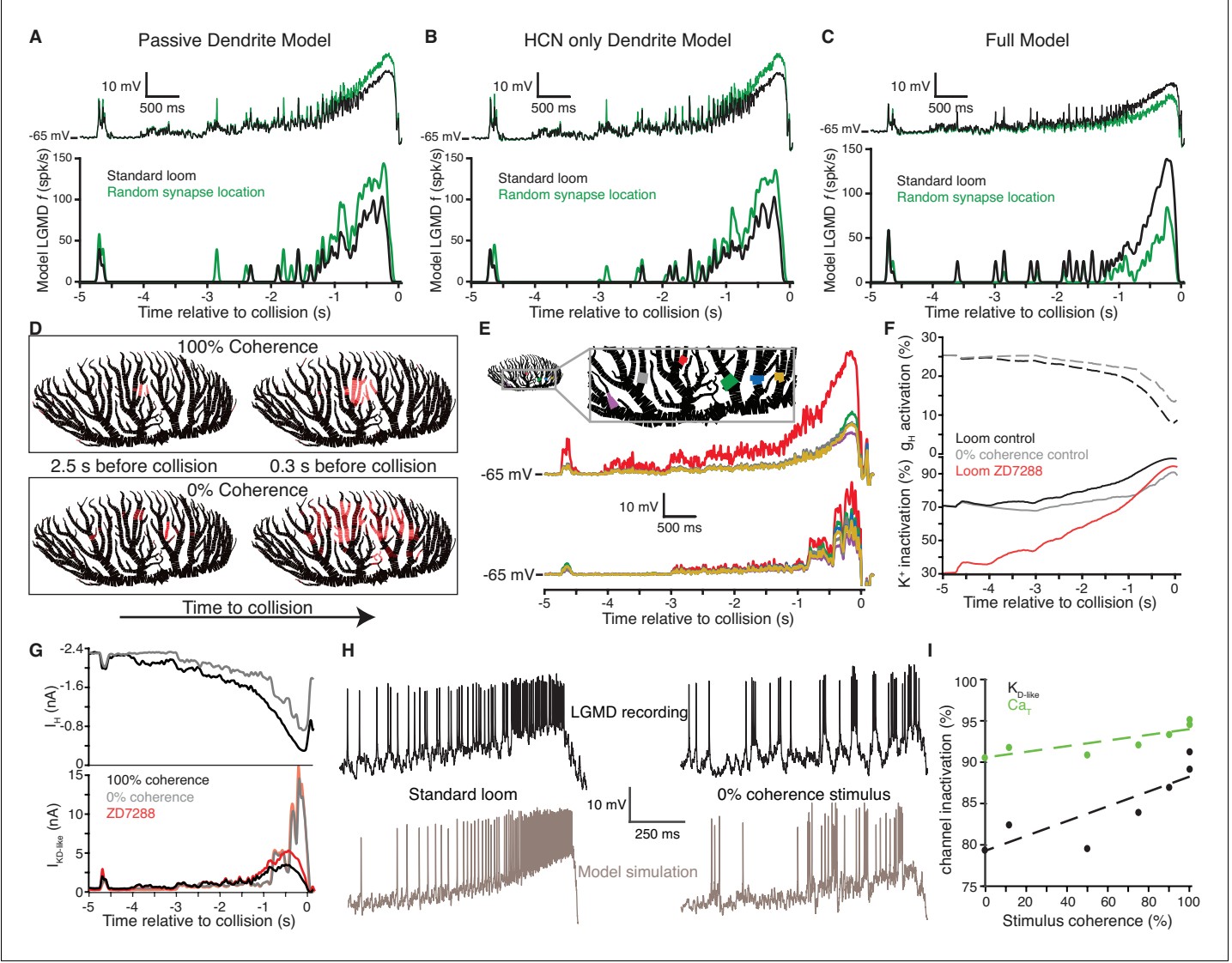

**Figure 7.** An active biophysical model reproduces the preference for coherent synaptic inputs. (**A**) A model LGMD with realistic morphology and passive dendrites generated a stronger response to spatially randomized inputs (green) than retinotopically arranged ones (black). Spatially randomizing inputs increased mean membrane potential 1.23 ± 0.01 mV at the base of field A (top) and increased firing by 59% (bottom). (**B**) Adding HCN channels to the dendrites did not change this trend. Spatially random inputs (green) increased mean membrane potential 1.23 ± 0.01 mV at the base of field A (top) compared to spatially coherent ones (black) and increased firing by 64% (bottom). (**C**) The full model, with both HCN and inactivating K$^+$ channels, however, had 1.25 ± 0.01 mV lower membrane potential at the base of field A and 61% less spiking in response to spatially random inputs. (**D**). Images of field A illustrating the branches receiving synaptic inputs. Brighter red indicates stronger inputs. During looming stimuli, excitation spreads slowly from a single location due to retinotopy (top). Spatially incoherent stimuli produce inputs spread over a much larger dendritic region. (**E**) The simulated membrane potential was measured at several dendritic locations indicated in the image at the top (traces below are color coded by location). For a coherent loom, a much larger depolarization occurs on dendritic branches receiving prolonged excitation. Incoherent stimuli generate a similar level of depolarization across the dendritic arbor. (**F**) At bottom, the time course of K$^+$ channel inactivation shows higher inactivation during a coherent looming stimulus (black) than an incoherent coarse stimulus (gray). After blocking HCN channels (red), the resting inactivation is much less and never reaches the inactivation level of control. At top, HCN channel activation is lower during coherent stimuli since K$^+$ channel inactivation leads to increased depolarization. (**G**) The time course of the HCN and K$_{D-like}$ total membrane currents during 100% and 0% coherence stimuli. I$_H$ decreases throughout both stimuli, but I$_{KD-like}$ increases more during incoherent stimuli. Red lines show the increased K$_{D-like}$ currents after HCN block. (**H**) Comparison of the membrane potential near the base of field A for experimental data (top) and model simulation (bottom) reveal a steady ramp up in firing rate in response to coherent looming stimuli and a burstier firing pattern in response to a 0% coherence coarse looming stimulus. (**I**) The mean channel inactivation during the last 2 s before collision increased with stimulus coherence for both K$^+$ and Ca$_T$ channels (with slopes of 0.094 and 0.030, respectively).

DOI: https://doi.org/10.7554/eLife.34238.039

*Figure 7 continued on next page*

*Figure 7 continued*

The following source data, source code and figure supplements are available for figure 7:

**Source code 1.** A Matlab script that will import the data in *Figure 7—source data 1* and generate the plots in *Figure 7*.
DOI: https://doi.org/10.7554/eLife.34238.041
**Source data 1.** An .xlsx spreadsheet with data plotted in *Figure 7*.
DOI: https://doi.org/10.7554/eLife.34238.042
**Figure supplement 1.** Additional comparisons of the LGMD model with experimental data.
DOI: https://doi.org/10.7554/eLife.34238.040

bursts (*Gabbiani and Krapp, 2006*; *Peron and Gabbiani, 2009*). In both experimental data and simulations, responses to spatially coherent stimuli generated more sustained, non-burst firing than transient burst firing (*Figure 7H*; *Figure 7—figure supplement 1F,G*). The model reproduced the trends in these data qualitatively rather than quantitatively (see Discussion). The decrease in bursting for coherent stimuli was also dependent on $Ca_T$ channel inactivation. Coherent stimuli produced a steady ramp up of membrane potential increasing $Ca_T$ inactivation, while incoherent stimuli produced more sudden depolarization, producing bursts. To investigate the role played in the smooth ramp up of activity during looming stimuli by dendritic $K_{D-like}$ inactivation vs. SIZ $Ca_T$ inactivation and the concomitant suppression of bursting, we plotted the two inactivation variables as a function of stimulus coherence. During the last 2 s before collision, when most firing occurred, the average inactivation of both $Ca_T$ and $K_{D-like}$ channels increased with stimulus coherence (*Figure 7I*). Yet, the slope of the best fit line for $K_{D-like}$ inactivation vs. stimulus coherence was three times as steep as that of $Ca_T$ inactivation. This confirms the relative importance of $K_{D-like}$ in coherence selectivity but also shows that interactions of multiple channels underlie the coherence selectivity of the LGMD model.

*Figure 8* illustrates the interactions of the channels involved in coherence selectivity during object approach. Dendritic field A receives retinotopic inputs across a compartmentalized arbor. The resting $g_H$ contributes to this compartmentalization by decreasing the electrical compactness and membrane time constant. In the model, the selectivity arises from fast negative feedback of $K_{D-like}$ activation embedded in the context of two slow positive feedbacks: one from $K_{D-like}$ inactivation and the other from $g_H$ deactivation. Spatially incoherent visual stimuli generate spatially dispersed synaptic inputs that depolarize many branches, rapidly increasing negative feedback by activation of $K_{D-like}$. This reduces its own slow inactivation and further depolarization generated by subsequent synaptic inputs (*Figure 6*; *Figure 6—figure supplement 1*). The spatially dispersed, transient excitation of incoherent stimuli generate transient depolarizations of the SIZ activating burst firing followed by $K_{Ca}$ activation that inhibits sustained spiking.

For spatially coherent stimuli, in contrast, the synaptic inputs continue to excite the same dendritic branches for a prolonged period. This prolonged local activation eventually causes $K_{D-like}$ inactivation and HCN deactivation, resulting in positive feedback by reducing $K^+$ current and increasing EPSP amplitude and summation. Additionally, the slow increase in depolarization propagates to the SIZ where it inactivates $Ca_T$ channels, reducing burst spiking and its subsequent negative feedback caused by $K_{Ca}$.

## Discussion

Here, we provide, to the best of our knowledge, the first demonstration of selectivity to spatial coherence for an ecologically important escape behavior (*Figure 1*). Our results suggest that this spatial discrimination relies upon discrimination of the broad spatial statistics of synaptic inputs within the dendrites of a single neuron. To examine this issue, we characterized active conductances and studied how HCN and inactivating $K^+$ channels produced selectivity for spatial coherence (*Figures 2*, *5* and *6*). Although our results suggest that spatial selectivity is in large part implemented within the LGMD's dendritic arbor, they do not rule out additional presynaptic mechanisms. Furthermore, we blocked HCN channels in freely moving animals, demonstrating that the selectivity of escape behavior depends on HCN channels enhancing spatially coherent responses (*Figure 3*).

Our experimental data suggest that HCN channels produce a selective enhancement for inputs generated by spatially coherent approaching objects. To the best of our knowledge, there are no previously described mechanisms by which ion channels could produce such spatial selectivity. While

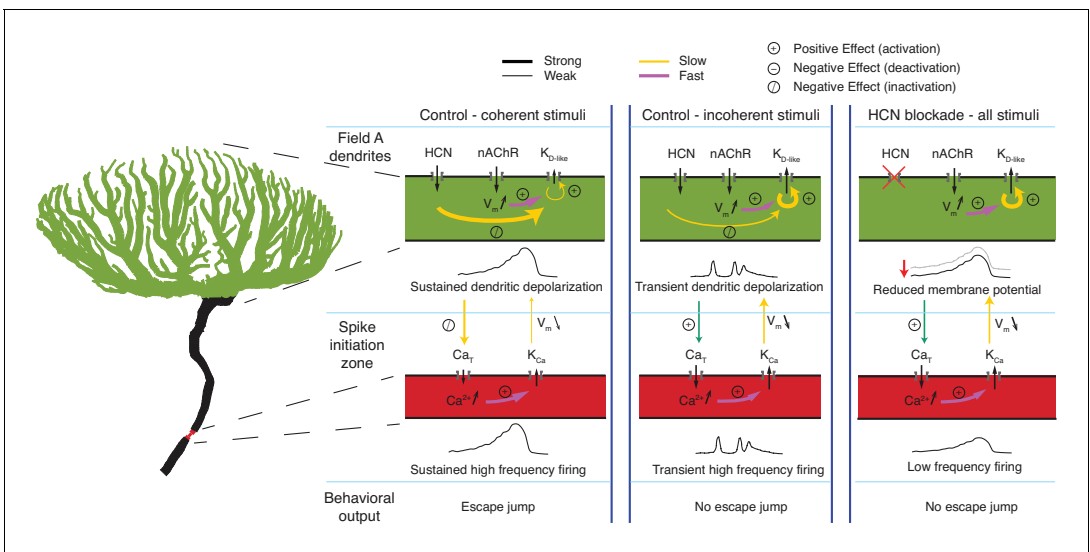

**Figure 8.** Schematic illustration of the effects of HCN channels on responses to looming stimuli. Arrows indicate current flow, changes in $Ca^{2+}$ or membrane potential, and the strongest interaction between channels, dendrite and SIZ, during looming stimulus detection. For all visual stimuli, excitatory nicotinic acetylcholine receptors (nAChR) produce a fast activation of $K_{D-like}$. In control conditions during coherent stimuli (left), the addition of $g_H$ on top of nAChR activation produces sustained depolarization which slowly inactivates $K_{D-like}$ channels within field A (green) and $Ca_T$ channels at the SIZ (red). This results in reduced $K_{D-like}$ and $K_{Ca}$ conductances and sustained high frequency firing. In control conditions during incoherent stimuli (middle), the increased dendritic area of nAChR activation increases the number of $K_{D-like}$ channels activated. The resulting increase in the hyperpolarizing $K_{D-like}$ conductance prevents the $g_H$-dependent inactivation of $K_{D-like}$ channels, producing only transient depolarization. The transient depolarization initiates $Ca_T$ driven bursts and subsequent $K_{Ca}$ conductance activation. This prevents the sustained high-frequency firing necessary for initiating escape. After $g_H$ blockade (right), the resting membrane potential is reduced, increasing the activatable $K_{D-like}$ and $Ca_T$ channels. Without $g_H$, depolarization cannot be sustained high enough to inactivate these channels leading to an increased $K_{D-like}$ and $K_{Ca}$ conductances and lower frequency firing which fails to produce escape behaviors.

DOI: https://doi.org/10.7554/eLife.34238.043

fast positive feedback from $Na^+$ channels, $Ca^{2+}$ channels or NMDA receptors can enhance the impact of clustered synaptic inputs (*Takahashi et al., 2012*; *Kleindienst et al., 2011*; *Poleg-Polsky and Diamond, 2016*; *Weber et al., 2016*; *Sivyer and Williams, 2013*; *Smith et al., 2013*; *Wilson et al., 2016*) it remains unclear whether they could also provide a way to select for broad spatiotemporal patterns of synaptic inputs. Based on biophysical modeling, we developed a plausible hypothesis explaining the underlying mechanisms, schematically illustrated in *Figure 8*. These mechanisms involve competition between depolarizing and hyperpolarizing conductances within a compartmentalized dendritic arbor, regulation of membrane potential to control levels of $K^+$ and $Ca^{2+}$ channel inactivation, and regulation of bursting.

The model illustrated in *Figure 8* was the simplest that reproduced the wide range of our experimental data. Detecting the differences in spatiotemporal patterns of synaptic inputs requires an electrotonically extended arbor. To test whether the dendritic morphology could be reduced without a loss of selectivity we compressed different dendritic regions into electrotonically equivalent cylinders. Despite containing the same conductances and the same passive properties as the full model, the simplified morphology markedly reduced the coherence selectivity (*Figure 7—figure supplement 1H*).

Although the specific kinetics and distributions of several channels in the LGMD remain to be characterized, the model is well grounded (Material and methods). After extensive searches through parameter space we could not find other combinations of mechanisms that reproduced the experimental data as well. Simulations were conducted with altered kinetics of HCN closing and $K_{D-like}$ inactivation, and both faster and slower kinetics reduced the coherence selectivity (*Figure 7—figure supplement 1I*).

Yet, the model did not reproduce quantitatively all our experimental results: for example, it underestimated the amount of transient firing at high coherence and, vice-versa, overestimated its

sustained firing (*Figure 7E*; *Figure 7—figure supplement 1F,G*, dashed lines). One likely reason is that details of the bursting mechanisms may be imperfectly tuned in the model, due to the absence of a second calcium-sensitive $K^+$ conductance (*Peron and Gabbiani, 2009*) or still uncharacterized properties of an M current (unpublished observations). Further confirmation of this will require future experimental tests of channel properties predicted by the model, including the precise location of HCN and $K_{D-like}$ channels within field A (*Figure 2F*), that $K_{D-like}$ and $Ca_T$ inactivate above $-70$ mV, and that $K_{D-like}$ inactivates slowly (in the range of 0.3–2 s).

HCN channels have long been known to influence dendritic integration in hippocampal pyramidal neurons (*Magee, 1998*), and $K_D$ as well (*Storm, 1988*). More recently, dendritic $K^+$ channels have been found to compartmentalize dendrites, and it has been suggested that spatiotemporal interactions between HCN and $K^+$ channels regulate neuronal excitability (*Harnett et al., 2013*; *Harnett et al., 2015*; *Mishra and Narayanan, 2015*). It is thus possible that selectivity for broad spatial synaptic input patterns arises in pyramidal neurons by mechanisms analogous to those described here. In thalamocortical neurons, HCN channels influence $K^+$ and $Ca^{2+}$ channel inactivation, thereby regulating bursting and excitability (*McCormick and Pape, 1990*; *McCormick, 1991*). HCN regulation of bursting has been tied to a rat model of absence epilepsy (*Ludwig et al., 2003*; *Kole et al., 2007*) and may also contribute to human epilepsy (*Bender et al., 2003*). In addition to possible disease states, HCN-dependent regulation of persistent or burst firing has also been involved in working memory (*Thuault et al., 2013*).

In summary, our results highlight how nonlinear dendritic conductances and their interactions confers the ability to reliably select synaptic patterns appropriate for the generation of visually guided escape behaviors. This highlights the computing power of individual neurons and may help design object segmentation algorithms for bio-inspired collision avoidance systems. As HCN conductances are ubiquitous, they may contribute to implement analogous computations in other species, including our own (*Bender et al., 2003*).

## Materials and methods

### Animals

All experiments were performed on adult grasshoppers 7–12 weeks of age (*Schistocerca americana*). Animals were reared in a crowded laboratory colony under 12 hr light/dark conditions. For experiments preference was given to larger females ~ 3 weeks after final molt that were alert and responsive. Animals were selected for health and size without randomization, and investigators were not blinded to experimental conditions. Sample sizes were not predetermined before experiments. For many experiments, a large number of experiments were conducted (e.g. >100 experiments in *Figure 2*). For technically difficult experiments (e.g. *Figure 3*), smaller sample sizes were used with enough replications to see a clear effect.

### Surgery

The surgical procedure for intracellular recordings was described previously (*Peron et al., 2009*; *Jones and Gabbiani, 2010*). For extracellular DCMD recordings in freely moving animals, we developed a novel chronic implant technique, allowing the same animals to be recorded over many days, based on previous methods (*Fotowat et al., 2011*). Grasshoppers were fixed ventral side up and a rectangular window was opened in their thorax. Air sacs were removed and the trachea were carefully separated to expose the ventral nerve cords. Two teflon-coated stainless steel wires 50 μm in diameter were cut to a length of ~4 cm and fashioned into hooks with the coating removed from the inside edge of the crook (supplier: California Fine Wire, Grover Beach, CA). The electrodes were implanted with the deinsulated region placed against the dorsomedial edge of the left nerve cord between the pro- and meso-thoracic ganglia. Slight tension was applied to the cord to maintain a fixed position against the wires, and the wires were set in place by waxing them to the left side of the thorax. The cuticle window was then closed and sealed with a wax-rosin mixture and Vetbond (3M, St. Paul, MN). A ground electrode made of the same wire as the hooks was placed outside on the thorax and embedded in the wax. All three wires were routed laterally and fixed to the dorsal pronotum using the wax-rosin mixture with just enough slack to allow normal pronotum movement. The ends of the wires were de-insulated and positioned pointing up to prevent the animal from

reaching them. After the surgery, animals were allowed a day to recover, and survived for up to 7 months during which time the animals behaved normally.

To connect the electrode wires to the amplifier during an experiment, the animals were held in place with transparent surgical tape (Dukal Corp, Ronkonkoma, NY). The free ends of the implanted electrodes were each attached to polyurethane-coated hook-up wire with a pair of gold-plated miniature connectors (0508 and 3061, Mill-Max, Oyster Bay, NY; wire diameter: 160 µm or 34 AWG, Belden, St. Louis, MO). The hook-up wires were braided together and loosely suspended directly above the animal to allow unrestrained movement. Neither the implantation surgery nor the connection of implanted wires to the amplifier caused a significant reduction in escape behavior (*Figure 3—figure supplement 1*).

## Visual stimuli

Visual stimuli presented during jump experiments were generated with custom software on a personal computer (PC) running the real-time operating system QNX 4 (QNX Software Systems), as previously described (*Gabbiani et al., 1999*). Identical visual stimuli for electrophysiological experiments were generated using Matlab and the PsychToolbox (PTB-3) on a PC running Windows XP. In both cases, a conventional cathode ray tube (CRT) monitor refreshed at 200 frames per second was used for stimulus display (LG Electronics, Seoul, Korea). Both monitors were calibrated to ensure linear, 6-bit resolution control over luminance levels. Visual stimuli were presented in blocks with each stimulus shown once per block and the order within the block randomized by the stimulus software for all experiments. For wide field stimuli presented to restrained animals, a 90–120 s delay was used between stimuli and grasshoppers were repeatedly brushed and exposed to light flashes and high frequency sounds to decrease habituation. Some animals still exhibited pronounced visual habituation (>50% reduction in peak firing rate from that animal's average response to the stimulus), and these data were excluded from analysis. In escape behavior experiments, a delay of at least 5 min (and usually ~15 min) was used between stimuli to prevent habituation. The drug effects were long lasting, so in all cases the control data was collected before the drug condition. Stimuli were randomly interleaved, no fatigue was evident within experimental conditions, and the drug effects reported are stimulus specific, so habituation or fatigue cannot explain the coherence-dependent results described. It cannot be ruled out that the exact change in firing is unaffected by habituation or fatigue, however.

Looming stimuli consisted of dark squares simulating the approach of a solid object on a collision course with the animal (*Hatsopoulos et al., 1995*). Briefly, the instantaneous angular size, $2\theta(t)$, subtended at one eye by a square of radius, $l$, approaching the animal at constant speed, $v$, is fully characterized by the ratio, $l/|v|$, since $\theta(t) = \tan^{-1}[l/(v\,t)]$. By convention, $v < 0$ for approaching stimuli and $t < 0$ before collision. Stimuli simulated approach with $l/|v|$ values of 50 or 80 ms from an initial subtended angle of 1.2° until filling the vertical axis of the screen (300 mm), lasting approximately 4 and 7 s for $l/|v| = 50$ and 80 ms, respectively. The maximum $2\theta$ values reached by the stimuli were either 136° or 80° for the freely behaving or restrained preparations, respectively, due to the differing distances of the eye to the screen.

'Coarse' looming stimuli were generated as in our earlier work (*Jones and Gabbiani, 2010*). Briefly, the stimulation monitor was first pixelated with a spatial resolution approximating that of the locust eye (2–3° x 2–3°), referred to as 'coarse' pixels. Each coarse pixel's luminance followed the same time course as that elicited by the edge of the simulated approaching object sweeping over its area. To alter the spatial coherence of these stimuli, a random two-dimensional Gaussian jitter with zero mean was added to each coarse pixel screen location. The jittered positions were rounded to the nearest available coarse pixel location on the screen to prevent any coarse pixels from overlapping. To control the amount of shifting and thus the resulting spatial coherence of the randomized stimulus, the standard deviation, σ, of the Gaussian was altered between 0° and a maximal angular value $\sigma_{max}$ determined by the procedure described in the next paragraph.

For a given Gaussian jitter σ, we determined the corresponding percent spatial coherence by averaging over 30 pseudo-random draws the minimal total angular distance that jittered coarse pixels had to be moved in each movie image to reconstitute the unaltered coarse looming stimulus. This distance was then normalized by the angular distance computed in the same way between a coarse loom and one with uniformly and independently drawn random spatial positions. Subtracting this normalized distance from one yields coherence values ranging from 100% when σ = 0% to 0%

when σ reaches a value $\sigma_{max}$ for which the jittered stimulus is indistinguishable from a totally random one. The value of $\sigma_{max}$ was different for freely behaving (80°) and restrained preparations (40°), due to the different distance between the screen and animal and thus the different angular expanse of the stimulus (see above).

For localized light flashes, a 1° x 1° luminance increase was presented briefly (~1 s) on a black background in the dark (*Figure 2—figure supplement 1A–C*). A window of 200 ms following the flash onset was used to quantify LGMD activity.

## Escape behavior

The behavioral experiments were conducted as previously described (*Fotowat and Gabbiani, 2007*). They were recorded with a high-speed digital video camera (GZL-CL-22C5M; Point Grey, Richmond, BC, Canada), equipped with a variable zoom lens (M6Z 1212–3S; Computar, Cary, NC). Image frames were recorded at 200 frames per second with the acquisition of each frame synchronized to the vertical refresh of the visual stimulation display (Xtium-CL PX4; Teledyne Dalsa, Waterloo Canada). Videos were made from the images and saved in lossless motion JPEG format using custom Matlab code. Measurements of the stimulus coherence's effect on escape behavior (*Figure 1F*) include a total of 202 trials from 66 animals with 1–9 trials per animal. Animals which did not jump in response to any stimuli were excluded from analysis, as done previously (*Fotowat and Gabbiani, 2007*).

## Electrophysiology

Electrophysiological experiments were performed as described previously (*Peron et al., 2009*; *Jones and Gabbiani, 2010*). Briefly, sharp-electrode LGMD intracellular recordings were carried out in both voltage-clamp and current-clamp modes using thin walled borosilicate glass pipettes (outer/inner diameter: 1.2/0.9 mm; WPI, Sarasota, FL). After amplification, intracellular signals were low-pass filtered (cutoff frequency: 10 kHz for $V_m$, and 5 kHz for $I_m$) and digitized at a sampling rate of at least 20 kHz.

We used a single electrode clamp amplifier capable of operating in discontinuous mode at high switching frequencies (typically ~25 kHz; SEC-10, NPI, Tamm, Germany). Responses to visual stimulation were measured in bridge mode, current injections were applied in discontinuous current clamp mode (DCC), and voltage-clamp recordings in discontinuous single-electrode voltage-clamp mode (dSEVC). All dSEVC electrodes had resistances < 15 MΩ. Electrode resistance and capacitance were fully compensated in the bath immediately prior to tissue penetration and capacitance compensation was readjusted after entering the neuron. If capacitance could not be fully compensated the recording was not used. In addition to previously described methods, a fluorescent dye (Alexa Fluor 594 hydrazide salt; Invitrogen, Thermo Fisher Scientific, Carlsbad, CA) was injected intracellularly and the cell was imaged with a CCD camera mounted to a stereomicroscope (GuppyPro F125B; Allied Vision Technologies, Exton, PA). This allowed subsequent visually guided positioning of the recording electrode. Within the LGMD, back-propagating action potentials (bAPs; measured from RMP to peak) decay as they spread into the dendrites. Data from dual recordings (unpublished) has revealed that the decay in bAPs is a better indicator of electrotonic distance than the path length, and it is also easier to reliably determine. So, we used bAP amplitude as the measure of electrotonic distance from the SIZ (*Figure 2E*).

During voltage clamp recordings, the membrane potential and current were measured simultaneously to ensure the desired membrane potential was maintained at the electrode location. The LGMD neuron is not electrotonically compact (*Peron et al., 2007*) and therefore the issue arises of how well its dendritic membrane potential is controlled through voltage-clamping at a single location ('space clamp'). The quality of the space clamp cannot be measured with a single electrode recording. In pyramidal neurons, the steady-state dendritic membrane potential is largely uncontrolled when voltage clamping originates at the soma (*Williams and Mitchell, 2008*). In contrast, in Purkinje cells, which have a dendritic structure more closely resembling that of the LGMD, the steady-state dendritic membrane potential is well controlled from the soma (*Roth and Häusser, 2001*). Simulations in NEURON (details below) were used to estimate the quality of the space clamp in the LGMD. For electrodes placed at the base of field A the average steady-state change in membrane potential within field A was 95% of the desired change (i.e. starting at rest, −65 mV, a −30 mV

step to $-95$ mV, yielded an average membrane potential across field A of $-65 + 0.95 \cdot (-30)$ or $-93.5$ mV). For electrodes placed further away from the base of field A, the quality of the space clamp decreased. Across the dendritic region used for voltage clamp recordings, the estimated quality of the space clamp ranged from 83-95% (average voltage command attenuation of 5-17%). We also carried out simulations as described in *Roth and Häusser (2001)* to assess the impact of these findings on the characterization of $g_H$ activation and kinetics. The effects were found to be mild, suggesting that the activation curve in *Figure 2F* might be slightly less steep and the time constants in Figure 2G slightly higher than if they were measured with perfectly space-clamped dendrites.

For characterizing $g_H$, 1–2 s hyperpolarizing current or voltage steps were injected in DCC or dSEVC mode, respectively, with 5 s between steps. Voltage clamp was needed to calculate the activation and time constant at a given membrane potential (*Figure 2G,H*), while current clamp was used for all other experiments because it allowed for easier to hold, longer lasting recordings. Different step amplitudes were randomly interleaved and at least six trials per step amplitude per animal were acquired. For each recording, we used at least four step amplitudes, with values selected to cover the activation range of $g_H$. Example recordings are shown in *Figure 2—figure supplement 1*. In most experiments, no holding current was applied between steps (held at the resting membrane potential), while in some experiments a positive holding current was applied (held near $-50$ mV) before hyperpolarizing steps and in other experiments a negative holding current was used (held near $-115$ mV) with depolarizing steps. Estimated activation curves (see below) were not different for recordings with different holding potentials, and the data were combined for analysis. Extracellular recordings were taken between the two hook electrodes on the nerve cord, differentially amplified and bandpass filtered from 100 to 5000 Hz (A-M Systems, model 1700, Carlsborg, WA). The amplitude of DCMD spikes was consistently the largest, allowing their identification with a simple threshold. DCMD spikes uniquely identify the LGMD neuron as they are in one-to-one correspondence with those of the LGMD (*O'Shea and Williams, 1974*).

Experiments with hyperpolarizing current during visual stimulation (*Figure 5—figure supplement 1*) were conducted by first staining the LGMD and then inserting an electrode near the base of field A. Within our LGMD model, the resting membrane current magnitude generated by HCN channels was ~2.5 nA. Experimentally, injecting –2.5 nA produced local hyperpolarization to $\leq -75$ mV, which is as hyperpolarized as any LGMD neuron became after HCN blockade (see *Figure 4A*). Visual stimuli of different coherences were presented either with zero current or –2.5 nA current injected from 20 s before stimulus onset through the end of the stimulus. Sets of single trials of each stimulus (randomized) were presented while alternating between 0 and –2.5 nA currents and continued until at least three trials of each stimulus were presented for both conditions.

## Pharmacology

Drugs were prepared in aqueous solution and mixed with physiological saline containing fast green (0.5%) to visually monitor the affected region. They were puffed using a pneumatic picopump (WPI, PV830, Sarasota, FL). For restrained experiments, injection pipettes had tip diameters of ~2 μm and were visually positioned with a micromanipulator against the posterior edge of the lobula, close enough that the ejected solution penetrated the optic lobe. Drugs were gradually applied while monitoring responses of the LGMD to visual inputs, and care was taken to prevent spread into presynaptic neuropils. Additionally, saline in the bath was exchanged immediately after puffing to prevent diffusion to other brain areas. We used drug concentrations of 10 mM for both ZD7288 and 4AP in the extracellular puff pipette. These concentrations were adjusted in pilot experiments to account for the low mobility of the drugs through the tissue in vivo, taking into account their approximate final concentration, as explained below.

Due to dilution of the drugs in the saline bath after puffing, the exact drug concentration at the level of the LGMD cannot be determined. However, our best estimate is ~200 μM for both ZD7288 and 4AP. This estimate comes from comparing the effect of the puffed drugs to those observed after bath application of the same drugs. For example, when bath applying ZD7288, the same level of blockade as from local puffing was achieved by adding 100 μl of 20 mM drug to ~5.5 ml of bath saline for a final concentration of ~350 μM. This concentration is an upper bound on the concentration at the level of the LGMD, since it lies ~150 μm deep within the optic lobe. For local puffing, less

than 1 µl of drug was used, which would generate a final bath concentration well below 1 µM after exchanging the saline in the bath, as explained above.

For intracellular application, the drug concentrations in the pipette were 1–5 mM for ZD7288 and 5 mM for 4AP. The final concentration inside the LGMD cannot be determined but is likely considerably lower, due to the large volume of the cell and the submicron diameter of the pipette. In those experiments, the effects of the drugs were comparable to those observed with extracellular application.

Although it cannot be known whether intracellularly applied ZD7288 or 4AP diffused out from within the LGMD, this seems highly unlikely to have affected our results. For example, the effects of intracellular ZD7288 application on the LGMD's membrane properties were consistent from a minute to an hour after application, giving no evidence of a slow diffusion across tissue that may have affected presynaptic sites. Further, the membrane effects on the LGMD were the same whether excitatory synaptic inputs were blocked with mecamylamine or not. To further rule out the possibility that the effects of ZD7288 observed during visual stimulation were caused by diffusion to presynaptic sites after intracellular application, we conducted visual stimulation experiments in which the $g_H$ conductance was blocked intracellularly with Cs$^+$ (*Figure 2—figure supplement 2*). Similar effects were seen on visual responses compared to ZD7288 application, although Cs$^+$ was not as specific a blocker since there was also evidence of partial block of K$^+$ conductances. For these experiments, a concentration of 150 mM CsCl was used in the recording pipette. We also attempted to block intracellularly the inactivating K$^+$ conductance by using 4-aminopyridine methiodide (4APMI) which is membrane impermeant (*Stephens et al., 1994*). 4APMI reacted strongly with the silver wire in the electrode forming AgI crystals, so a platinum wire was used for the experiments. Unfortunately, 4APMI which is larger than 4AP failed to block the inactivating K$^+$ conductance even at recording pipette concentrations as high as 50 mM. Nonetheless, presynaptic effects are unlikely as we never observed increases in spontaneous EPSPs within the LGMD following 4AP application and the presynaptic neurons have no information about the overall spatial pattern of the stimulus. In all there were no indications of any nonspecific drug effects on presynaptic neurons that might have influenced visual responses.

To observe the effects of ZD7288 in freely moving animals, stereotaxic injections were made through a hole in the dorsal rim region of the right eye. The animal was restrained and the head was placed in a small clamp attached to a 3-axis micromanipulator (Narishige, Tokyo, Japan). Head tilt was positioned manually by fixing the animal at the pronotum. After the head was precisely positioned, a ~ 0.5 mm hole was made through the dorsal end of the eye with a steel probe. A drop of saline solution was placed covering the hole to prevent drying or coagulation of the hemolymph. A glass pipette with a tip diameter of 1–2 µm and a taper length >2 mm from shoulder to tip was positioned with a Leica (Wetzlar, Germany) manual micromanipulator and lowered just above the eye. The ZD7288 solution (2 mM in saline with 1% fast green) was puffed into the saline drop covering the dorsal rim to determine the appropriate air pressure ejection level. The saline droplet was immediately removed and replaced to prevent spread of ZD7288 to photoreceptors. Next, the pipette was lowered through the eye along the dorsal rim of the optic lobe to the lobula (~1.5 mm) while enough positive pressure was maintained to prevent clogging. In control experiments, LGMD activity was measured before and after penetration of the pipette in the lobula to ensure that visual inputs were not damaged by the procedure (*Figure 3—figure supplement 1C*). Ejection volume was estimated from monitoring changes in the meniscus position of the saline within the visible region of the pipette. After pressure ejection of ZD7288, the pipette was removed and checked for clogs or breaks. The hole in the eye was sealed with a small amount of Vetbond (3M, St. Paul, MN), carefully ensuring that no glue spread onto the rest of the eye.

Following the conclusion of the experiment, the animal was euthanized and the head was dissected (~2 hr post injection). Fast green staining was used to confirm that the solution was injected into the lobula (*Figure 3—figure supplement 1D*). In initial experiments, bath application of ZD7288 was found to reduce visual responses as did application of ZD7288 directly to photoreceptors. When puffing ZD7288 within the lobula, however, even if the solution occasionally spread to the medulla or lamina, visual responses remained similar to those observed after intracellular application. This suggests that there are likely HCN channels within the photoreceptor layer, as is the case in mammals (*Barrow and Wu, 2009*), but that any HCN channels within the medulla and lamina (*Hu et al., 2015*) do not influence LGMD inputs under our experimental conditions. Because ZD7288

was applied extracellularly, it may have affected other descending neurons whose processes are located in the immediate vicinity of the LGMD dendrites. This is, however, unlikely to have affected escape behaviors, since decrease in escape was tightly correlated with a reduction of LGMD firing rate determined in independent experiments (*Figure 2N*). In addition, earlier selective ablation experiments have shown that under our experimental conditions nearly all escape behaviors depend solely on LGMD firing (*Fotowat et al., 2011*).

## Data analysis and statistics

Data analysis was carried out with custom MATLAB code (MathWorks, Natick, MA). Linear fits were based on Pearson's linear correlation coefficient, denoted by 'r' in figure legends, with corresponding p values testing significant differences from zero. Non-linear fits, including the activation curve and time constant in *Figure 2* and all exponential fits described below were made with the Matlab function 'lsqcurvefit', which minimizes the least square error between the data and fitting function. Goodness of fit was denoted by $R^2$, calculated as one minus the sum squared error of the fit divided by the sum square deviation from the mean of the data. For behavioral experiments and the comparison of membrane depolarization with stimulus angular distance (*Figure 6* and *Figure 6—figure supplement 1*), individual trials were used as independent sample points for statistical tests. In all other cases, individual trials were averaged and these trial averages were used for statistical tests.

The sag amplitude was measured as the difference in membrane potential between the peak hyperpolarization during a current step and the steady-state value at the end of the step. The sag time constant was calculated from fitting a single exponential to the membrane potential for the period starting 15 ms after peak hyperpolarization to the end of the current step (*Figure 2—figure supplement 1C*). The hyperpolarizing step currents were also used for calculating membrane time constants. The membrane time constant was calculated by fitting a single exponential to the membrane potential for the period from 0.5 to 13 ms after the start of hyperpolarizing current injection.

The fitted activation curve of the HCN conductance was based on a Boltzmann equation reflected along the vertical axis to produce decreasing $g_H$ with increasing v:

$$g_H(v) = \frac{g_{max}}{1 + e^{\frac{v - v_{1/2}}{s}}}.$$

The steady-state conductance, $g_H$, is a function of the membrane potential, v, depending on three parameters: the maximum conductance, $g_{max}$, the half-activation potential, $v_{1/2}$, and the steepness, s. The parameters were fitted from voltage-clamp data based on the equation

$$g_H(v_2) - g_H(v_1) = \Delta I_H / (v_2 - E_H),$$

where $v_1$ and $v_2$ are the starting and ending clamp potentials and $E_H$ is the reversal potential of the HCN conductance, −35 mV, used for all animals. $\Delta I_H$ is the experimentally measured change in membrane current produced by the voltage step after transients have settled. $\Delta I_H$ was measured by fitting a single exponential to the current time-course 15 ms after the step onset and up to its end (*Figure 2—figure supplement 1D*). This period captured the slow change in clamp current due to $g_H$ and offered clear experimental advantages over other estimations methods. As all experiments were done in vivo, it was not feasible to reliably block other putative voltage-gated channels. Hence, the most reliable measurements of $\Delta I_H$ were obtained at hyperpolarized membrane potentials where other active conductances can be safely discounted. Voltage clamping the LGMD to depolarized potentials where all HCN channels will be closed (> −40 mV) was not technically feasible, and the use of tail currents yielded less reliable measurements due to contamination by other active conductances.

The time constant of the HCN conductance ($\tau_H$) was fit using a function symmetric with respect to its maximum, $\tau_{max}$,

$$\tau_H(v) = \frac{\tau_{max}}{0.5\left(e^{\frac{v - v_{1/2}}{s}} + e^{\frac{v_{1/2} - v}{s}}\right)} + \tau_{min}.$$

Here, $v_{1/2}$ is the membrane potential with the slowest activation, s is the steepness, and $\tau_{min}$ the

minimum activation time. Fitted points were obtained from the single exponential fits to $I_H$ for both hyperpolarizing (channel opening) and depolarizing (channel closing) voltage steps.

Comparisons of sag amplitudes were obtained with current steps yielding a peak hyperpolarization of ~105 mV (*Figure 2D–F*). For *Figure 2D,E*, all values to steps within the range of −95 to −115 mV were pooled. For *Figure 2F* interpolation of values at nearby potentials was used to estimate sag amplitude at −105 mV to have a single common value for all recordings. Statistical comparisons between sag measurements in different subcompartments of the LGMD (*Figure 2D,E*) were carried out using a Kruskall-Wallis analysis of variance (KW) corrected for multiple comparisons with Tukey's Honestly Significant Difference Procedure (KW-MC). To determine the correct statistical test for comparison, we used a Lillifors test of normality (alpha = 0.20) and comparison of equality of variance. Much of the data was non-normally distributed and variances increased after drug application so most comparisons were made using the Wilcoxon rank sum test (WRS) which does not assume normality or equality of variance. For displaying non-normal data, average values were given as median and variance was displayed as median average deviation (mad). Mean and standard deviation were used for normally distributed values, as indicated in figure legends. Before carrying out paired tests, we determined if the paired differences where normally distributed. The changes in slope reported in *Figure 6D,E* were the only non-normal paired difference, so for this we used the non-parametric signed-rank test. Percent activation at rest (*Figure 2I*) was calculated through bootstrapped activation curves from current clamp data. Unpaired t statistics were calculated from the bootstrapped mean and variance of activation at the resting membrane potential (−65 mV) (*Efron and Tibshirani, 1993*).

Simulated excitatory postsynaptic potentials (sEPSPs) were generated by injecting a series of five current waveforms with a set delay between them. Each waveform, *I(t)*, had a time course resembling that of an excitatory synaptic current,

$$I(t) = A\left(1 - e^{-t/\tau_1}\right)e^{1-t/\tau_2}$$

with peak amplitude *A*, rising time constant $\tau_1$ = 0.3 ms, and falling time constant $\tau_2$ = 3.0 ms. Summation was calculated as the ratio $(p_5-p_1)/p_1$, with $p_1$ and $p_5$ being the peak amplitude of the membrane potential relative to rest during the 1st and 5th sEPSP. In *Figure 4F*, we plotted the integrated membrane potential (relative to rest) divided by the integrated input current (charge) giving a value in units of mV ms/nA ms=M Ω that is readily comparable to input resistance.

Spike counts elicited by looming stimuli were measured from the start of the stimulus until the time of expected collision, and peak firing rates were calculated by convolving the spike rasters with a 20 ms sd Gaussian as has been done in previous studies (*Hatsopoulos et al., 1995*; *Gabbiani et al., 1999*; *Peron et al., 2009*; *Jones and Gabbiani, 2010*; *Fotowat and Gabbiani, 2007*). After statistics were conducted on unnormalized data, firing rates were normalized before averaging across animals to reduce the between animal variability in responses. This allowed for the clearest comparison of the role of stimulus coherence across conditions and with the biophysical model of the LGMD. Normalized firing rates (*Figure 2M* and *Figure 5C*) were calculated by dividing the response amplitude for each stimulus by that animal's maximal response amplitude under control conditions (insets of *Figures 2M* and *5C*). Dividing by the maximum response was chosen to show that 100% coherent stimuli generated the maximum response and to give an easy indication of the amount of change from a standard/fully coherent looming response. These individually normalized rates were then averaged across animals. The relative change in response due to a drug (*Figure 5D*) was calculated by dividing the difference in response between control and drug conditions by the drug condition response. This produced percentages covering similar ranges, and so allowed for the best comparison and graphical illustration of their relative effects.

'Sustained firing' was defined as the longest period in which the instantaneous firing frequency remained above a 20 spk/s threshold. For each trial, the number of spikes within this longest period was considered the 'sustained response' and all spikes outside of this period were counted as the 'transient response' (*Figure 7—figure supplement 1F,G*). These 'sustained' and 'transient' measures were used instead of 'burst' and 'non-burst' statistics based on interspike intervals because the LGMD can generate sustained high frequency firing with similar interspike intervals within and outside of bursts.

To compare changes in membrane potential and stimulus angular distance (*Figure 6*; *Figure 6—figure supplement 1*), we identified newly changing coarse pixels in a specified stimulus frame from those that had begun to darken from their background luminance value in earlier frames ('earlier changing'). We then computed the mean minimal distance of newly changing coarse pixels with respect to earlier changing ones. In parallel, changes in the membrane potential were averaged from 25 ms following the appearance of the newly changing coarse pixels until a new group of pixels began to darken. More precisely, we identified six time periods during the stimuli when the luminance of newly changing coarse pixels is decreasing for over 50 ms and they typically have mean angular distances larger than 1° from earlier changing ones. For these six different time periods during each trial, we calculated the linear correlation between these mean angular distances and membrane depolarizations, as explained above. Early in the stimulus presentation, there are fewer coarse pixels changing luminance and less resulting depolarization. To better illustrate the relationship between these variables, the angular distances and membrane potentials were normalized independently for each of the six time windows. The normalization consisted of subtracting the minimum control value and then dividing by the range for control data within each time window. The unnormalized data and example stimulus frames from all time periods are shown in *Figure 6—figure supplement 1*.

To compare jump probabilities between saline- and ZD7288-injected animals (*Figure 3A*), we computed 95% bootstrap confidence intervals of the population mean in each condition with the help of the built-in Matlab function 'bootci' (using the bias corrected and accelerated method). If there was no overlap of the 95% confidence intervals, the groups were considered significantly different. The reported p-values for these comparisons were the 'achieved significance level' (ASL) statistic for two-sample testing of equality of means with unequal variance (Algorithm 16.2 in *Efron and Tibshirani, 1993* ).

Coherence selectivity was calculated as the slope of the relationship between stimulus coherence and spike count and is reported in units of spikes per percent coherence. For control experiments, the median correlation coefficient of this relationship between stimulus coherence and spike count was 0.97, making the regression slope a reliable indicator of the selectivity.

For box plots, the center line shows the median, the upper and lower box limits mark the 25th and 75th percentile of the distribution, and the 'whiskers' above and below each box extend 1.5 times the interquartile range up to the minimum and maximum values. Points beyond the whiskers mark outliers. Notches, when present, have a width of 1.57 times the interquartile range divided by the square root of the number of data points.

## Neuronal modeling

To better understand the mechanisms of the LGMD's remarkable coherence selectivity, we developed a detailed biophysical model using the NEURON simulation environment. We employed the parallel version of NEURON and a Rice University supercomputing cluster for extensive parameter sweeps and simulations. Three-dimensional reconstructions of the LGMD's dendritic morphology were obtained from two-photon scans using the software suite Vaa3D (vaa3d.org). The resulting model contained 2518 compartments, 1266 of which belonged to dendritic field A.

To reproduce the active properties of the LGMD several voltage-gated channel types were included. Some of them had been used in previous simulations (*Peron et al., 2007*; *Jones and Gabbiani, 2012*), including the fast $Na^+$ and delayed rectifier $K^+$ ($K_{DR}$) channels generating action potentials. $K_{DR}$ channels were distributed throughout the cell, but dendritic branches contained no fast $Na^+$ channels as supralinear summation is never seen in LGMD dendrites. HCN channels had kinetics matching experimental data (*Figure 2*) and were placed in dendritic field A with density increasing towards the distal dendritic endings. Inactivating $K^+$ channels ($K_{D-like}$) were also distributed throughout field A with density increasing toward distal endings. A slow non-inactivating $K^+$ channel (M) was distributed throughout the axon, the spike initiation zone (SIZ), and the main neurite connecting the dendritic subfields to the SIZ. Its peak density was at the SIZ. Additionally, low-threshold $Ca^{2+}$ ($Ca_T$) and $Ca^{2+}$-dependent $K^+$ ($K_{Ca}$) channels were placed at the SIZ and on half of the neurite connecting the SIZ and the dendritic subfields, matching results from our earlier work (*Peron and Gabbiani, 2009*).

Effective modeling often relies on keeping things as simple as possible, so we initially tested a previous LGMD model (*Jones and Gabbiani, 2012*) with additional HCN channels matching

experimental kinetics (*Figure 2*) added to the dendrites. When this failed to reproduce any coherence selectivity, $K_{D-like}$ channels were added. Then we added a complex dendritic morphology, complex presynaptic transforms, and additional active conductances to the model. While a wide range of parameters of this more complex model reproduced responses to current injection data, only a narrow parameter regime was found that reproduced the roles of $g_H$ and $K_{D-like}$ in the spatial coherence preference. The resulting model and mechanistic explanation (*Figure 7 and 8*), while quite complex, is still the simplest model that reproduced the wide range of LGMD responses tested.

To test the impact of the elaborate dendritic structure of field A on coherence selectivity, we simplified its electrotonic structure in successive steps using Rall's law of electrotonically equivalent cylinders (*Rall, 1959*). This was done by iterating through dendritic branches by selecting dendritic segments according to their electrotonic distance from the base of field A in steps of 0.04 times the dendritic space constant ($\lambda$). An equivalent compartment was created from the group of dendritic segments found at each successive electrotonic distance from the base of field A. For the least reduction (the 'six branch' case shown in *Figure 7—figure supplement 1*), the grouping was limited to dendritic segments that shared a common connection with the base of field A. For further reduction, the group of selected compartments was expanded to segments with adjacent connections to the base of field A. The equivalent compartment size was set by Rall's equivalent 3/2 diameter law. Each channel's density was set to the surface area weighted mean of its density in the selected dendritic segments, and all synaptic inputs to these segments were transferred. Tests of channel kinetics were run using 10-fold changes to the time constant of HCN activation and $K_{D-like}$ inactivation. For the 'fast' kinetics, the maximal HCN channel activation was set to $\tau_{max}$ = 135 ms and the $K_{D-like}$ inactivation to $\tau_{max}$ = 105 ms. For the 'slow' kinetics, the maximal HCN channel activation was $\tau_{max}$ = 13.5 s and the $K_{D-like}$ inactivation was $\tau_{max}$ = 10.5 s. The values used for all other simulations in the manuscript were 1.35 s and 1.05 s, respectively.

Evaluation of how well this model informs about the actual neural processes requires some review of the experimental data to which it was constrained. The strength and timing of synaptic inputs was generated based on single facet stimulation data (*Jones and Gabbiani, 2010*). Excitatory synaptic input locations were based on the retinotopy and synaptic overlap determined by functional imaging (*Zhu and Gabbiani, 2016*; *Peron et al., 2009*). The time course of synaptic inputs was based on experiments stimulating individual facets (*Jones and Gabbiani, 2010*) and the pattern of depolarization measured during the current experiments. The presence of standard Hodgkin-Huxley $Na^+$ and $K^+$ currents was assumed, HCN and $K_{D-like}$ channels were based on the current work, the $Ca_T$ and $K_{Ca}$ channels were based on *Peron and Gabbiani, 2009*, the M current was based on our own currently unpublished findings. For each of these channels, conductance and kinetic parameters were adjusted to match experimental data with firing frequency vs. injected current curves and spike waveform used to tune fast $Na^+$ and $K^+$ channels, changes in input resistance and resting membrane potential after pharmacological blockade used to adjust HCN, $K_{D-like}$, and M parameters, while $Ca_T$ and $K_{Ca}$ were adjusted to match intrinsic burst (currently unpublished) and spike frequency adaptation data (*Peron and Gabbiani, 2009*). Channel distributions were similarly grounded in experimental data, when available, and were manually fit to find working parameters.

To estimate space clamp quality (see Materials and methods: Electrophysiology), we used the Impedance object class in NEURON and measured the percent voltage attenuation from an electrode location to each compartment within field A of the model. The average attenuation was calculated by weighting each section by its surface area to calculate the average change of membrane potential within field A.

## Data and code availability

The full model and simulation code are available in the public repository ModelDB, accession number 195666. Experimental data and code used to generate figures are available as Source Data and Code.

## Acknowledgements

We would like to thank Y Zhu and E Sung for the two-photon LGMD scan and reconstruction, respectively, and S Peron and J Reimer for feedback on an earlier draft of the manuscript. FG would like to thank the Marine Biological Laboratory (Woods Hole, MA) for a 2006 Faculty Summer

Research Fellowship that helped initiate this line of research. This work was supported by grants from NIH (MH-065339) and NSF (DMS-1120952), as well as by a NEI Core Grant for Vision Research (EY-002520–37). This work was also supported in part by the Cyberinfrastructure for Computational Research, funded by NSF under Grant CNS-0821727 and Rice University.

## Additional information

### Funding

| Funder | Grant reference number | Author |
| --- | --- | --- |
| National Institutes of Health | MH065339 | Fabrizio Gabbiani |
| National Science Foundation | DMS-1120952 | Fabrizio Gabbiani |
| National Science Foundation | IIS-1607518 | Fabrizio Gabbiani |
| National Eye Institute | Core Grant for Vision Research EY-002520–37 | Fabrizio Gabbiani |

The funders had no role in study design, data collection and interpretation, or the decision to submit the work for publication.

### Author contributions

Richard Burkett Dewell, Data curation, Software, Validation, Investigation, Visualization, Methodology, Writing—original draft, Writing—review and editing; Fabrizio Gabbiani, Conceptualization, Resources, Software, Supervision, Funding acquisition, Methodology, Writing—original draft, Project administration, Writing—review and editing

### Author ORCIDs

Richard Burkett Dewell ⓘ https://orcid.org/0000-0003-2430-8184

Fabrizio Gabbiani ⓘ http://orcid.org/0000-0003-4966-3027

### Decision letter and Author response

Decision letter https://doi.org/10.7554/eLife.34238.054
Author response https://doi.org/10.7554/eLife.34238.055

## Additional files

### Supplementary files

• Source code 1. ErrorBars.m matlab file.
DOI: https://doi.org/10.7554/eLife.34238.044

• Source code 2. ErrorPlot.m matlab file.
DOI: https://doi.org/10.7554/eLife.34238.045

• Source code 3. Isvect.m matlab file.
DOI: https://doi.org/10.7554/eLife.34238.046

• Source code 4. lin_Fit.m matlab file.
DOI: https://doi.org/10.7554/eLife.34238.047

• Source code 5. linear_plotter.m matlab file.
DOI: https://doi.org/10.7554/eLife.34238.048

• Supplementary file 1. Statistics table. A table including additional details on the statistical analyses presented in the Results and Figures.
DOI: https://doi.org/10.7554/eLife.34238.049

• Transparent reporting form
DOI: https://doi.org/10.7554/eLife.34238.050

## Data availability

The full model and simulation code are available in the public repository ModelDB, accession number 195666. Experimental data and code used to generate figures are available as Source Data and Code.

The following dataset was generated:

| Author(s) | Year | Dataset title | Dataset URL | Database and Identifier |
|---|---|---|---|---|
| Richard Burkett Dewell, Fabrizio Gabbiani | 2017 | LGMD with 3D morphology and active dendrites | http://modeldb.yale.edu/195666 | ModelDB, 195666 |

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
