## [Decision Letter]

Thank you for submitting your article "Biophysics of object segmentation in a collision-detecting neuron" for consideration by *eLife*. Your article has been favorably evaluated by Eve Marder (Senior Editor) and three reviewers, one of whom, Fred Rieke (Reviewer #1), is a member of our Board of Reviewing Editors.

The reviewers have discussed the reviews with one another and the Reviewing Editor has drafted this decision to help you prepare a revised submission.

All three reviewers agreed that the paper described a very interesting result and were enthusiastic about the approach. We also agreed that the paper could be strengthened in several ways. Four specific points follow; these and other points are detailed in the individual reviews.

1) Utility of the model. The model presented is quite complex and does not lead to a clear intuitive insight into how the interaction between HCN and KD type channels gives rise to looming sensitivity. The paper would benefit from more intuition about what the key properties are (time course, voltage dependence, etc.) that control interactions between these channels and give rise to looming sensitivity.

2) Inclusion of "standard" looming stimuli in analysis. It was unclear whether the standard looming stimuli were included in the fits in Figure 2J and 4C. This point would appear to impact the fits considerably.

3) Statistical tests. The paper is lacking statistical tests in several places. This includes analysis of significance for key results (a partial list includes the cAMP results and the slopes of lines in Figure 2E – but generally this should be addressed throughout the paper).

4) Division between main and supplementary figures. Some key pieces of data show up only in the supplementary figures (e.g., the pharmacology in Supplementary Figure 1). Supplementary Figure 4 is also quite central to the paper. The paper would benefit from inclusion of any key results in the main figures.

Reviewer #1:

This paper describes the basis of sensitivity of looming sensitivity in grasshopper LGMD neurons. The breadth of the paper – from single cell conductances to behavior – is impressive and the paper provides a rare link between the mechanistic basis of dendritic computation, neural responses and behavior. The general significance of the work is set up nicely in the Introduction. There are several issues that should be strengthened.

Abstract: The Abstract creates the simple expectation that HCN channels are going to explain looming sensitivity. Only later in the paper is it clear that it is the interaction of HCN channels and a depolarization-activated K channel. It would help to make this clear from the start.

HCN channel block. The pharmacology in Supplementary Figure 1 is quite central to the paper. I think it should also be included as a main figure. It also at present is restricted to example cells, and needs some population analysis. Further it is not clear why some experiments were in current clamp and others in voltage clamp.

Statistical tests. There are a number of effects for which significance should be evaluated. Some examples are: (1) the cAMP results in Figure 2F; (2) the slopes of the black lines in Supplementary Figure 4; (3) the slope in Figure 2E.

Figure 4 relies heavily on Supplementary Figure 4. Further, Supplementary Figure 4 is quite confusing. I think this entire analysis needs to be described more clearly and more of it needs to be included in the main text.

Reviewer #2:

In this manuscript, Dewell and Gabbiani examine molecular mechanisms underlying loom detection in the LGMD neuron in locust. The use a clever stimulus in which they can continuously modulate the coherence of a loom-like stimulus to investigate how different channels affect the coherence tuning of the LGMD response. First, they show that LGMD neurons show a sag under hyperpolarization that is cAMP-modulated and is eliminated by an antagonist to HCN channels. Using this antagonist, they show various properties of LGMD responses that depend on HCN channels, but show that these properties alone do not account for the coherence tuning of the neurons. This antagonist also reduces the jump rate for fully coherent stimuli. They use a second antagonist to show that potassium channels are also involved in coherence tuning, but show somewhat opposing effects to the HCN channels. Finally, the authors conclude with a realistic model of LGMD firing, including many channels and compartments, and show that after tuning parameters, they reproduce some of their data with this model, importantly reproducing the primary effects of the HCN and potassium channels on the loom coherence tuning curves.

The coherence knob on the loom stimuli is clever and a useful way to interrogate these mechanisms, and I found the pharmacology and measurements all mostly convincing and well-presented, with a few small points enumerated below. My major concern was with the modeling. I've read through these sections three or four times now, and it's still not clear to me what is going on. Some of this speaks to presentation, and I think this could be improved. But one question is: what is the purpose of the model? A detailed model with thousands of compartments and many fitted parameters can be used to determine whether you're missing any crucial components in principle, by asking whether the sophisticated model can reproduce results with verisimilitude? But such models typically won't give an intuition for what's going on. I want a model that gives me intuition about the biophysical processes involved. In the supplement, the authors say that this is the simplest model that reproduces the results, but surely there must be a toy model with fewer and more simply arranged spatial compartments that would give intuition for some specific coherence results, even if it didn't reproduce things exactly. In part, Figure 7 gave me pause because this coherence/incoherence is necessarily a spatiotemporal integration issue, as the authors emphasize in the introduction, but the schematic has a single spatial compartment. To really get what's going on, we need at least two compartments. Or a one-dimensional chain of compartments with an edge sweeping across, which could show currents/activations/voltages at different time-points during the coherent vs. incoherent edge? I'd really like to have a better intuition for what is happening, and what timescales matter for the HCN activity and its inactivation of the KD-like channels.

Reviewer #3:

Dewell and Gabbiani use a novel visual 'looming' stimuli of varying spatial coherence. With the aid of pharmacological intervention, the authors explore mechanisms underlying response characteristics of looming neurons (and escape behaviors) in the grasshopper. Here they provide evidence that spatiotemporal patterning on the LGMD neuron's dendrites induced by looming stimuli, elicit responses via HCN and other channel dynamics. Whether and how neurons use broad spatial components of the dendritic input statistics to compute information (rather than via presynaptic mechanisms) is an important question in the neurosciences of interest to a general audience.

1) I have an important question with respect to data analysis. In the critical Figure 2J, the authors chose to include a standard loom in addition to a 100% coherence loom (effectively two data points at 100%), though the reasoning for this is not provided. Does the line of best fit (red) include the standard loom ('star') data point at 100% coherence? If so, why would this 'standard loom' stimulus composed of very different frequency components be added to this analysis? The primary hypothesis of this manuscript depends on the slope of this line to increasing coherence (coherence preference) and this selectivity looks to be largely affected by the presence of this additional, confounding data (standard vs. 100% coarse).

What is the coherence preference for ZD7288 with this point removed and is it still significantly different than control (Figure 2K)? If not, why not? If the slopes of these lines are not significantly different, then the parsimonious explanation for the data set is that the ZD7288 is causing something akin to the hyperpolarizing injection of current (Supplementary Figure 3D). Testing for a significant difference between the coherence preference for ZDZ7288 (without standard loom) with that obtained for the.-2.5nA current injection would make the author's point more convincing. In fact for ease of comparison, coherence preferences for all conditions (control, ZD7288, 4AP, Cs+, -2.5dc) could be presented in a single figure (with error bars and tests for significance). The authors may then like to develop hypotheses with respect to response differences in standard loom compared to 100% coherence coarse loom induced by HCN blockade.

Note that a similar 'extra' data point seems to also be having a very large impact on the 4AP result (Figure 4C).

Additionally, as the primary interpretation is from the slope rather than the overall strength of these lines of best fit (Figure 2J), the reader should be provided (a) the analysis window used for spike count (b) the y-intercept (0% coherence) relationship to any spontaneous activity and (c) why a single control data point was used as the normalization (% max). To further complicate matters, the individual data underlying Figure 2J presented in Supplementary Figure 3A strangely shows a weighted line (with data points) for one single example, rather than the average. It is in fact this Supplementary Figure 3A (the non-highly derived) that makes the authors point much more convincing, with most red lines looking flat – however the authors should again separate standard vs. 100% coarse loom to be fully validated.

For their main hypothesis to be supported, I believe the authors should start again with Figure 2J and 2K. Calculating the mean, spike counts (from Supplementary Figure 3A), without including the standard zoom. If a normalizing factor must be used, one less dependent on a single value would be better (e.g. mean across all coherences).

Following, the rest of the manuscript flows very well, delving further into proposed mechanisms underlying this preference. This includes solid proposals of interactions between HCN and voltage-gated K^+^ channels, and robust modeling efforts.

2) Figure 2I reveals interesting time courses of control and ZD7288 that are not discussed in further depth. With intracellular application of ZD7288 the spiking activity (red dashed line) ends earlier than control. This is not simply a gain scaling, as response onset initially builds with the same time course as the control. These waveforms are also different with the extracellular applications.

How do these differences in time course affect further analysis? That is, what is the time window considered for calculating spike count? Do the model simulations explain any of these interesting temporal dynamics? Furthermore, if peak spike rates were used instead, would the calculated coherence preference across conditions exhibit similar traits?

3) It is not clear in the main text when individual trials (across animals) are considered as independent samples for the statistical tests. Or if and when such replicates are averaged into a single sample.

4) Although data in Figure 2K is presented as median and the authors applied a paired t-test (similarly in other tests). However, there is not a solid description of the ordering of these paired interventions. I presume the solutions could not be washed out, therefore all these results are with controls first, followed by treatments. Although the authors cannot get around this difficulty, it should be described clearly so that readers can consider confounding factors such as animal fatigue and neuronal habituation on such responses.

[Editors' note: further revisions were requested prior to acceptance, as described below.]

Thank you for resubmitting your work entitled "Biophysics of object segmentation in a collision-detecting neuron" for further consideration at *eLife*. Your revised article has been favorably evaluated by Eve Marder as the Senior Editor, and Fred Rieke as the Reviewing Editor. The original reviewers were consulted but were not asked for formal reviews.

This is a revision of an interesting paper on dendritic computation in a visually-sensitive neuron in grasshopper. The paper has improved in revision, but a few issues remain. These all center around making the paper maximally accessible to a broad audience. Most generally, the model is still difficult, and I would encourage the authors to take another pass through that section to see if it can be clarified further (some specific suggestions are below).

Abstract, penultimate sentence: This sentence could get broken into two for clarity.

Subsection HCN channels in dendritic field A are implicated in coherence tuning”, end of first paragraph: This is where you first introduce HCN channels as a possible mechanism for selective responses to coherent motion. You could expand the proposal here a bit, as it is not immediately clear how HCN channels would serve this function (and it later becomes clear that they do not by themselves). An interested reader will be trying to develop a conceptual framework at this point in the paper, and likely will be confused about the proposal. A similar issue comes up in the last paragraph of the aforementioned subsection and in the first paragraph of the subsection “HCN channels mediate coherence tuning of escape behaviors”. In all these cases a reader thinking about how HCN would mediate the observed effects will likely be confused. The intuition for the role of HCN comes in the subsection “HCN channels affect membrane properties and synaptic summation” – I think this should come earlier.

"A similar result held[…]" – not sure you mean similar here, as that most naturally would mean larger and faster from previous sentence.

Subsection “HCN channels in dendritic field A are implicated in coherence tuning” (and thereafter): I think an implicit assumption is that the current you measure is dominated by g_H_. It would be good to state this assumption explicitly.

Subsection “Compartmental modeling highlights role of K^+^ and Ca^2+^ channel inactivation in coherence tuning”, fourth paragraph: this paragraph is tough because of the back and forth between activation and inactivation. Is there a simple way to illustrate the interplay between voltage, HCN channel activity and KD inactivation? Maybe with activation and inactivation curves for KD and the range of voltages explored with and without HCN channels?

Figure 7D: it would be helpful to describe more fully here why full inactivation is important, as opposed for example to modulation of the level inactivation (which looks larger with HCN channels blocked).

Figure 6D/E: Some statistics are needed here.

Subsection “Compartmental modeling highlights role of K^+^ and Ca^2+^ channel inactivation in coherence tuning”, fourth paragraph: reference to Figure 7C seems out of place here.

Figure 7G: Can you do these same calculations with HCN blocked?

Subsection “Compartmental modeling highlights role of K^+^ and Ca^2+^ channel inactivation in coherence tuning”, last paragraph: I think ending the results with the transient calcium channel part weakens the paper a bit since that is the last thing a reader walks away with, and that is the least established part of the paper.

Discussion, second paragraph: I think sodium or calcium channels could provide spatial selectivity – if you agree this sentence should get modified.

"increases in bursting.[…] " is awkward.

Discussion: there is a good deal of text in the Discussion that goes back over the model and how multiple channel types act together. I think this could get consolidated in the Results, and made briefer in the Discussion.

---

## [Author Response]

All three reviewers agreed that the paper described a very interesting result and were enthusiastic about the approach. We also agreed that the paper could be strengthened in several ways. Four specific points follow; these and other points are detailed in the individual reviews.

We would like to thank the reviewers for their criticism. We have strived to address them as comprehensively as possible and believe that the manuscript has been improved as a result. We provide immediately below a brief summary of the changes for the four specific points, with more details in response to the individual reviews.

1) Utility of the model. The model presented is quite complex and does not lead to a clear intuitive insight into how the interaction between HCN and KD type channels gives rise to looming sensitivity. The paper would benefit from more intuition about what the key properties are (time course, voltage dependence, etc.) that control interactions between these channels and give rise to looming sensitivity.

We have carried out additional simulations, rewritten the modeling section and modified the corresponding figures to improve the presentation of the model.

2) Inclusion of "standard" looming stimuli in analysis. It was unclear whether the standard looming stimuli were included in the fits in Figure 2J and 4C. This point would appear to impact the fits considerably.

The “standard” looming stimuli were included in the fits. We had not thought of this caveat. After repeating the analysis excluding the “standard” looming stimuli we find that the results are nearly unchanged. The revised manuscript excludes the “standard” looming stimuli from all fits.

3) Statistical tests. The paper is lacking statistical tests in several places. This includes analysis of significance for key results (a partial list includes the cAMP results and the slopes of lines in Figure 2E – but generally this should be addressed throughout the paper).

We went systematically over the results and added statistical test as requested.

4) Division between main and supplementary figures. Some key pieces of data show up only in the supplementary figures (e.g. the pharmacology in Supplementary Figure 1). Supplementary Figure 4 is also quite central to the paper. The paper would benefit from inclusion of any key results in the main figures.

We have modified the main figures to include additional key results as suggested.

Reviewer #1:This paper describes the basis of sensitivity of looming sensitivity in grasshopper LGMD neurons. The breadth of the paper – from single cell conductances to behavior – is impressive and the paper provides a rare link between the mechanistic basis of dendritic computation, neural responses and behavior. The general significance of the work is set up nicely in the Introduction. There are several issues that should be strengthened.Abstract: The Abstract creates the simple expectation that HCN channels are going to explain looming sensitivity. Only later in the paper is it clear that it is the interaction of HCN channels and a depolarization-activated K channel. It would help to make this clear from the start.

Thank you, we have added a reference to inactivating K channels in the Abstract.

HCN channel block. The pharmacology in Supplementary Figure 1 is quite central to the paper. I think it should also be included as a main figure. It also at present is restricted to example cells, and needs some population analysis. Further it is not clear why some experiments were in current clamp and others in voltage clamp.

We have added a panel to Figure 2 (new panel B) illustrating the sag and the effect of ZD7288 on it. In addition, we added population averages summarizing the effect of ZD7288 on sag amplitude and time constant to Figure 2D and E. Cs+ population data has been added to Figure 2—figure supplement 1.

Current clamp was used whenever possible because it is technically easier to implement. Voltage clamp was used to measure the channel kinetics illustrated in Figure 2G and H. A statement to this effect was added to the Materials and methods (subsection “Electrophysiology”, fourth paragraph).

Statistical tests. There are a number of effects for which significance should be evaluated. Some examples are: (1) the cAMP results in Figure 2F; (2) the slopes of the black lines in Supplementary Figure 4; (3) the slope in Figure 2E.

Statistical tests have been added throughout, following the indications of the reviewers.

Figure 4 relies heavily on Supplementary Figure 4. Further, Supplementary Figure 4 is quite confusing. I think this entire analysis needs to be described more clearly and more of it needs to be included in the main text.

To address this point, we have moved several panels of the Figure 4—figure supplement 1 to the main text. In addition, we have added new illustrations of the data analysis procedure and an expanded description (subsection “K^+^ channels complement HCN channels in generating coherence tuning”, last paragraph) to the clarify the results. The old Figure 4 has thus been converted into two new figures in the revised manuscript, Figures 5 and 6.

Reviewer #2:[…] The coherence knob on the loom stimuli is clever and a useful way to interrogate these mechanisms, and I found the pharmacology and measurements all mostly convincing and well-presented, with a few small points enumerated below. My major concern was with the modeling. I've read through these sections three or four times now, and it's still not clear to me what is going on. Some of this speaks to presentation, and I think this could be improved. But one question is: what is the purpose of the model? A detailed model with thousands of compartments and many fitted parameters can be used to determine whether you're missing any crucial components in principle, by asking whether the sophisticated model can reproduce results with verisimilitude? But such models typically won't give an intuition for what's going on. I want a model that gives me intuition about the biophysical processes involved. In the supplement, the authors say that this is the simplest model that reproduces the results, but surely there must be a toy model with fewer and more simply arranged spatial compartments that would give intuition for some specific coherence results, even if it didn't reproduce things exactly. In part, Figure 7 gave me pause because this coherence/incoherence is necessarily a spatiotemporal integration issue, as the authors emphasize in the introduction, but the schematic has a single spatial compartment. To really get what's going on, we need at least two compartments. Or a one-dimensional chain of compartments with an edge sweeping across, which could show currents/activations/voltages at different time-points during the coherent vs. incoherent edge? I'd really like to have a better intuition for what is happening, and what timescales matter for the HCN activity and its inactivation of the KD-like channels.

We have comprehensively restructured the modeling section so as give a better intuition of the main components of the model and the role they play. We hope that the main ingredients of the model and how it works will now be clear.

Initially, we started with much simpler models than the one presented here in the hope of finding a compact explanation to the phenomenon. However, these models never worked properly. It is only when we added an extended dendritic tree with a corresponding large number of synaptic inputs distributed over it that we could reproduce the experimental results. This is due to the fact that the extended dendritic tree allows sufficient compartmentalization for the spatially different synaptic inputs generated by coherent and incoherent stimuli to have distinct effects on the local activation and inactivation of the H and K_D-like_conductances. It is possible that a simplified model explaining the phenomena exists, but we have not yet been able to design one. Simply collapsing the electrotonic structure of the model and localizing synaptic inputs in a few compartments fails, see the newly added Figure 7—figure supplement 1. We think the task of designing a simplified model attempting to reproduce the results is better left for future work.

It was not our intention in Figure 8 (previously Figure 7) to suggest that the channel schematics we used are confined to single dendritic compartments. On the contrary, it has to occur over the extended spatial dendritic tree. We have modified the figure to make this clearer.

Reviewer #3:[…] 1) I have an important question with respect to data analysis. In the critical Figure 2J, the authors chose to include a standard loom in addition to a 100% coherence loom (effectively two data points at 100%), though the reasoning for this is not provided. Does the line of best fit (red) include the standard loom ('star') data point at 100% coherence? If so, why would this 'standard loom' stimulus composed of very different frequency components be added to this analysis? The primary hypothesis of this manuscript depends on the slope of this line to increasing coherence (coherence preference) and this selectivity looks to be largely affected by the presence of this additional, confounding data (standard vs. 100% coarse).

We did not think about this potential caveat and thank the reviewer for pointing it out. We have thus reanalyzed the data excluding the standard looms and all the results hold without any changes. The results reported in the revised manuscript omit the standard looming data from their analyses.

What is the coherence preference for ZD7288 with this point removed and is it still significantly different than control (Figure 2K)? If not, why not? If the slopes of these lines are not significantly different, then the parsimonious explanation for the data set is that the ZD7288 is causing something akin to the hyperpolarizing injection of current (Supplementary Figure 3D). Testing for a significant difference between the coherence preference for ZDZ7288 (without standard loom) with that obtained for the.-2.5nA current injection would make the author's point more convincing. In fact for ease of comparison, coherence preferences for all conditions (control, ZD7288, 4AP, Cs+, -2.5dc) could be presented in a single figure (with error bars and tests for significance). The authors may then like to develop hypotheses with respect to response differences in standard loom compared to 100% coherence coarse loom induced by HCN blockade.

The coherence preference is unchanged when the standard loom data is removed. Following the reviewer’s suggestion, we have plotted changes for the main pharmacological manipulations together in Figure 5D to illustrate the different effect that each manipulation has on firing rate changes as a function of stimulus coherence. We would like to emphasize that we never found any substantial difference between the responses to coarse and standard looming stimuli, either in this paper or in an earlier one where they were extensively studied as well, but in a different context (Jones and Gabbiani, 2010).

Note that a similar 'extra' data point seems to also be having a very large impact on the 4AP result (Figure 4C).

We assume the reviewer is referring to the change in standard loom response after ZD7288, as 4AP had nearly identical effects on coarse and standard loom. We have now excluded the standard looming data points from the analysis of changes for both drugs (Figure 5D of the revised manuscript) and this produced no change in the results.

Additionally, as the primary interpretation is from the slope rather than the overall strength of these lines of best fit (Figure 2J), the reader should be provided (a) the analysis window used for spike count (b) the y-intercept (0% coherence) relationship to any spontaneous activity and (c) why a single control data point was used as the normalization (% max). To further complicate matters, the individual data underlying Figure 2J presented in Supplementary Figure 3A strangely shows a weighted line (with data points) for one single example, rather than the average. It is in fact this Supplementary Figure 3A (the non-highly derived) that makes the authors point much more convincing, with most red lines looking flat – however the authors should again separate standard vs. 100% coarse loom to be fully validated.

To address this point and the following one, we have added Supplementary Figure 3 of the previous manuscript to Figure 2 of the revised manuscript. It is now Figure 2K.

In the experiments reported in Figure 2K-M (previously Supplementary Figure 3 and Figure 2J, K), the answer to the questions raised above are as follows.

a) We computed the spike count over the entire trial (no post experiment analysis window selection was ever carried out).

b) There was no spontaneous activity of the LGMD.

c) A single point was used for normalization to allow easy comparison of experimental and model data relative to coherent loom responses

We have added a sentence in the revised manuscript explaining those points (subsection “HCN channels in dendritic field A are implicated in coherence tuning”, fourth paragraph).

To address point (c), in addition to adding Supplementary Figure 3 to the main text, we have plotted the mean data and carried out statistical significance tests for the slopes.

For their main hypothesis to be supported, I believe the authors should start again with Figure 2J and 2K. Calculating the mean, spike counts (from Supplementary Figure 3A), without including the standard zoom. If a normalizing factor must be used, one less dependent on a single value would be better (e.g. mean across all coherences).

We have carried out the suggested data analysis, thank you. Results are reported in Figure 2K and legend. We kept the normalization to the coherent looming response in Figure 2M, since the new Figure 2K addresses the point and this allows a direct comparison with the model data.

Following, the rest of the manuscript flows very well, delving further into proposed mechanisms underlying this preference. This includes solid proposals of interactions between HCN and voltage-gated K^+^ channels, and robust modeling efforts.2) Figure 2I reveals interesting time courses of control and ZD7288 that are not discussed in further depth. With intracellular application of ZD7288 the spiking activity (red dashed line) ends earlier than control. This is not simply a gain scaling, as response onset initially builds with the same time course as the control. These waveforms are also different with the extracellular applications.How do these differences in time course affect further analysis? That is, what is the time window considered for calculating spike count? Do the model simulations explain any of these interesting temporal dynamics? Furthermore, if peak spike rates were used instead, would the calculated coherence preference across conditions exhibit similar traits?

We did not dwell in the differences in firing rate time course, although we agree that there may be interesting points to be characterized there in future work. We note, however, that some of the differences between the raw spiking traces and the firing rates are due to smoothing with the Gaussian window, and that noise would have to be factored into the data analysis to see if the differences in extracellular responses are significant. In any case, these differences do not affect the results because we did not select a time window to analyze spiking, but rather used the spikes elicited over the entire trial (see above). We do not know if the model can explain the dynamics differences although this is again something we will consider in future work. As for the peak spike rates, as illustrated in Figure 1G, peak rate indeed decreases with coherence. We have now also added plots to Figure 2—figure supplement 2 showing that calculating coherence preference from the peak spike rates yields very similar results.

3) It is not clear in the main text when individual trials (across animals) are considered as independent samples for the statistical tests. Or if and when such replicates are averaged into a single sample.

We have modified the text to make this clear.

4) Although data in Figure 2K is presented as median and the authors applied a paired t-test (similarly in other tests). However, there is not a solid description of the ordering of these paired interventions. I presume the solutions could not be washed out, therefore all these results are with controls first, followed by treatments. Although the authors cannot get around this difficulty, it should be described clearly so that readers can consider confounding factors such as animal fatigue and neuronal habituation on such responses.

The within condition variability across animals in Figure 2L (previously 2k) was not normal, which is why the median and mad where chosen to illustrate population averages. The change caused by ZD7288 application, however, was normally distributed. Which is why the paired t-test was used. Note, however, that the corresponding non-parametric test yielded the same results (p = 0.002; signed-rank test). We have modified the Materials and methods section to make this clear (subsection “Data analysis and statistics”, sixth paragraph). We have also updated the Materials and methods section (subsection “Visual stimuli”, first paragraph) to address the ordering of drug conditions and the possible fatigue confound.

[Editors' note: further revisions were requested prior to acceptance, as described below.]

This is a revision of an interesting paper on dendritic computation in a visually-sensitive neuron in grasshopper. The paper has improved in revision, but a few issues remain. These all center around making the paper maximally accessible to a broad audience. Most generally, the model is still difficult, and I would encourage the authors to take another pass through that section to see if it can be clarified further (some specific suggestions are below).

We have made another pass on the section, implemented as best as possible the suggestions below, and think that the description was improved.

Abstract, penultimate sentence: This sentence could get broken into two for clarity.

It is now two sentences, and hopefully clearer.

Subsection HCN channels in dendritic field A are implicated in coherence tuning”, end of first paragraph: This is where you first introduce HCN channels as a possible mechanism for selective responses to coherent motion. You could expand the proposal here a bit, as it is not immediately clear how HCN channels would serve this function (and it later becomes clear that they do not by themselves). An interested reader will be trying to develop a conceptual framework at this point in the paper, and likely will be confused about the proposal. A similar issue comes up in the last paragraph of the aforementioned subsection and in the first paragraph of the subsection “HCN channels mediate coherence tuning of escape behaviors”. In all these cases a reader thinking about how HCN would mediate the observed effects will likely be confused. The intuition for the role of HCN comes in the subsection “HCN channels affect membrane properties and synaptic summation” – I think this should come earlier.

We appreciate the suggestion. We added several sentences providing an intuitive explanation of the proposed role of HCN channels to follow (subsection “HCN channels in dendritic field A are implicated in coherence tuning”, first paragraph.)

"A similar result held[…]" – not sure you mean similar here, as that most naturally would mean larger and faster from previous sentence.

This sentence was rewritten to make it clearer.

Subsection “HCN channels in dendritic field A are implicated in coherence tuning” (and thereafter): I think an implicit assumption is that the current you measure is dominated by g_H_. It would be good to state this assumption explicitly.

We have added an explicit mention of why we believe the current measured is dominated by g_H_ (subsection “HCN channels in dendritic field A are implicated in coherence tuning”, third paragraph)

Subsection “Compartmental modeling highlights role of K^+^ and Ca^2+^ channel inactivation in coherence tuning”, fourth paragraph: this paragraph is tough because of the back and forth between activation and inactivation. Is there a simple way to illustrate the interplay between voltage, HCN channel activity and KD inactivation? Maybe with activation and inactivation curves for KD and the range of voltages explored with and without HCN channels?

This paragraph was rewritten to reduce confusion, and hopefully now is clearer. Unfortunately, we have not found a simple illustration to show the interactions, as the difference is not only a question of the voltage level or range. It relies more on the temporal pattern of excitatory inputs received on the dendrites. Both stimuli generate enough depolarization for inactivation, but in only one case is that level maintained for long enough. Hopefully these points are now clear from the text.

Figure 7D: it would be helpful to describe more fully here why full inactivation is important, as opposed for example to modulation of the level inactivation (which looks larger with HCN channels blocked).

The reference should have been to Figure 7F in the previous submission. We have corrected this mistake (subsection “Compartmental modeling highlights role of K^+^ Ca^2+^ channel inactivation in

coherence tuning.”, fourth paragraph). Full inactivation is not inherently important in the model, and that sentence was reworded to remove the statement. It is the level of inactivation that is important, and specifically the level of inactivation in the branches receiving the inputs.

Figure 6D/E: Some statistics are needed here.

We have now added the statistics in the figure legend.

Subsection “Compartmental modeling highlights role of K^+^ and Ca^2+^ channel inactivation in coherence tuning”, fourth paragraph: reference to Figure 7C seems out of place here.

This sentence was changed to reference the traces in 7E showing the time course of the membrane potential.

Figure 7G: Can you do these same calculations with HCN blocked?

We added red lines to 7G showing the same calculation after HCN block and modified the main text accordingly.

Subsection “Compartmental modeling highlights role of K^+^ and Ca^2+^ channel inactivation in coherence tuning”, last paragraph: I think ending the results with the transient calcium channel part weakens the paper a bit since that is the last thing a reader walks away with, and that is the least established part of the paper.

To address this point and the last one about the discussion raised below, we have moved the description of the model Figure 8 to the end of the Results.

Discussion, second paragraph: I think sodium or calcium channels could provide spatial selectivity – if you agree this sentence should get modified.

The ions passed by the channel are likely not critical, but the time course and sensitivity to changes in membrane potential near rest likely are. The currently described dendritic Na^+^ and Ca^2+^ channels appear too fast to provide the slower and broader selectivity described here. An additional sentence was added to make this point (Discussion, second paragraph)

"increases in bursting[…]" is awkward.

This sentence was rewritten.

Discussion: there is a good deal of text in the Discussion that goes back over the model and how multiple channel types act together. I think this could get consolidated in the Results, and made briefer in the Discussion.

As explained above, much of the model description was removed from the discussion and incorporated into the results.